# Vulnerability to climate change of United States marine mammal stocks in the western North Atlantic, Gulf of Mexico, and Caribbean

Matthew D. Lettrich[1]*, Michael J. Asaro[2], Diane L. Borggaard[3], Dorothy M. Dick[4], Roger B. Griffis[5], Jenny A. Litz[6], Christopher D. Orphanides[2], Debra L. Palka[2], Melissa S. Soldevilla[6], Brian Balmer[7], Samuel Chavez[8], Danielle Cholewiak[2], Diane Claridge[9], Ruth Y. Ewing[6], Kristi L. Fazioli[10], Dagmar Fertl[11], Erin M. Fougeres[12], Damon Gannon[13], Lance Garrison[6], James Gilbert[14], Annie Gorgone[15], Aleta Hohn[16], Stacey Horstman[12], Beth Josephson[8], Robert D. Kenney[17], Jeremy J. Kiszka[18], Katherine Maze-Foley[19], Wayne McFee[20], Keith D. Mullin[21], Kimberly Murray[2], Daniel E. Pendleton[22], Jooke Robbins[23], Jason J. Roberts[24], Grisel Rodriguez- Ferrer[25], Errol I. Ronje[26], Patricia E. Rosel[27], Todd Speakman[28], Joy E. Stanistreet[29], Tara Stevens[30], Megan Stolen[31], Reny Tyson Moore[32], Nicole L. Vollmer[33], Randall Wells[32], Heidi R. Whitehead[34], Amy Whitt[35]

**1** ECS Under Contract for Office of Science and Technology, NOAA Fisheries, Silver Spring, Maryland, United States of America, **2** Northeast Fisheries Science Center, NOAA Fisheries, Woods Hole, Massachusetts, United States of America, **3** Greater Atlantic Regional Fisheries Office, NOAA Fisheries, Gloucester, Massachusetts, United States of America, **4** Office of Protected Resources, NOAA Fisheries, Silver Spring, Maryland, United States of America, **5** Office of Science and Technology, NOAA Fisheries, Silver Spring, Maryland, United States of America, **6** Marine Mammal and Turtle Division, Southeast Fisheries Science Center, NOAA Fisheries, Miami, Florida, United States of America, **7** Dolphin Relief and Research, Clancy, Montana, United States of America, **8** Integrated Statistics, Woods Hole, Massachusetts, United States of America, **9** Bahamas Marine Mammal Research Organisation, Marsh Harbour, Abaco, Bahamas, **10** Environmental Institute of Houston, University of Houston - Clear Lake, Houston, Texas, United States of America, **11** Ziphius EcoServices, Magnolia, Texas, United States of America, **12** Southeast Regional Office, NOAA Fisheries, Saint Petersburg, Florida, United States of America, **13** University of Georgia Marine Institute, Sapelo Island, Georgia, United States of America, **14** University of Maine, Orono, Maine, United States of America, **15** CIMAS, University of Miami, Under Contract for NOAA Fisheries Southeast Fisheries Science Center, Beaufort, North Carolina, United States of America, **16** Marine Mammal and Turtle Division, Southeast Fisheries Science Center, NOAA Fisheries, Beaufort, North Carolina, United States of America, **17** Graduate School of Oceanography, University of Rhode Island, Narragansett, Rhode Island, United States of America, **18** Department of Biological Sciences, Institute of Environment, Florida International University, Miami, Florida, United States of America, **19** CIMAS, University of Miami, Under Contract for Marine Mammal and Turtle Division, Southeast Fisheries Science Center, NOAA Fisheries, Miami, Florida, United States of America, **20** National Centers for Coastal Ocean Science, National Ocean Service, National Oceanic and Atmospheric Administration, Charleston, South Carolina, United States of America, **21** Marine Mammal and Turtle Division, Southeast Fisheries Science Center, NOAA Fisheries, Pascagoula, Mississippi, United States of America, **22** Anderson Cabot Center for Ocean Life at the New England Aquarium, Boston, Massachusetts, United States of America, **23** Center for Coastal Studies, Provincetown, Massachusetts, United States of America, **24** Marine Geospatial Ecology Lab, Duke University, Durham, North Carolina, United States of America, **25** University of Puerto Rico Mayaguez, Mayaguez, Puerto Rico, United States of America, **26** National Centers for Environmental Information, NOAA, Stennis Space Center, Hancock County, Mississippi, United States of America, **27** Marine Mammal and Turtle Division, Southeast Fisheries Science Center, NOAA Fisheries, Lafayette, Louisiana, United States of America, **28** National Marine Mammal Foundation, Charleston, South Carolina, United States of America, **29** Fisheries and Oceans Canada, Dartmouth, Nova Scotia, Canada, **30** CSA Ocean Sciences, East Greenwich, Rhode Island, United States of America, **31** Blue World Research Institute, Merritt Island, Florida, United States of America, **32** Sarasota Dolphin Research Program, Chicago Zoological Society, Sarasota, Florida, United States of America, **33** CIMAS, University of Miami, Under Contract for Marine Mammal and Turtle Division, Southeast Fisheries Science Center, NOAA Fisheries, Lafayette, Louisiana, United States of America, **34** Texas Marine Mammal Stranding Network, Galveston, Texas, United States of America, **35** Azura Consulting, Garland, Texas, United States of America



**Data Availability Statement:** All relevant data are within the paper and its Supporting Information files.

**Funding:** This project was funded by the NOAA Fisheries Office of Science and Technology. ECS Federal, Inc in support of NOAA NMFS Office of Science and Technology provided salary for author MDL. This research was carried out [in part] under the auspices of the Cooperative Institute for Marine and Atmospheric Studies (CIMAS), a Cooperative Institute of the University of Miami and the National Oceanic and Atmospheric Administration, cooperative agreement # NA20OAR4320472 (AG, KMF, NLV). The funders had no role in study design, data collection and analysis, decision to publish, or preparation of the manuscript.

**Competing interests:** The authors have declared that no competing interests exist.

* matthew.lettrich@noaa.gov

# Abstract

Climate change and climate variability are affecting marine mammal species and these impacts are projected to continue in the coming decades. Vulnerability assessments provide a framework for evaluating climate impacts over a broad range of species using currently available information. We conducted a trait-based climate vulnerability assessment using expert elicitation for 108 marine mammal stocks and stock groups in the western North Atlantic, Gulf of Mexico, and Caribbean Sea. Our approach combined the exposure (projected change in environmental conditions) and sensitivity (ability to tolerate and adapt to changing conditions) of marine mammal stocks to estimate vulnerability to climate change, and categorize stocks with a vulnerability index. The climate vulnerability score was very high for 44% (n = 47) of these stocks, high for 29% (n = 31), moderate for 20% (n = 22), and low for 7% (n = 8). The majority of stocks (n = 78; 72%) scored very high exposure, whereas 24% (n = 26) scored high, and 4% (n = 4) scored moderate. The sensitivity score was very high for 33% (n = 36) of these stocks, high for 18% (n = 19), moderate for 34% (n = 37), and low for 15% (n = 16). Vulnerability results were summarized for stocks in five taxonomic groups: pinnipeds (n = 4; 25% high, 75% moderate), mysticetes (n = 7; 29% very high, 57% high, 14% moderate), ziphiids (n = 8; 13% very high, 50% high, 38% moderate), delphinids (n = 84; 52% very high, 23% high, 15% moderate, 10% low), and other odontocetes (n = 5; 60% high, 40% moderate). Factors including temperature, ocean pH, and dissolved oxygen were the primary drivers of high climate exposure, with effects mediated through prey and habitat parameters. We quantified sources of uncertainty by bootstrapping vulnerability scores, conducting leave-one-out analyses of individual attributes and individual scorers, and through scoring data quality for each attribute. These results provide information for researchers, managers, and the public on marine mammal responses to climate change to enhance the development of more effective marine mammal management, restoration, and conservation activities that address current and future environmental variation and biological responses due to climate change.

## Introduction

Global climate change is altering the physical conditions that support inshore, coastal, and oceanic ecosystems [1–4]. The increasing levels of heat and carbon dioxide in the atmosphere translate to increasing ocean temperatures, rising sea levels, decreasing dissolved oxygen, declining sea ice coverage, and ocean acidification [5–9]. The additional heat also drives shifting patterns in precipitation [10, 11], salinity [12], and ocean circulation [13–16]. Some of these changes in environmental and habitat conditions are similarly impacting different regions. For example, relative sea level rise in waters of the southeastern United States, Gulf of Mexico (GOMx), and Caribbean Sea has been accelerating [6, 17–20]. However, for other parameters, the observed and predicted rates and magnitudes of these changes vary regionally. In another example, sea surface temperature (SST) [21] and ocean acidity [22–24] have

increased rapidly in the western North Atlantic (WNA), while dissolved oxygen levels throughout the northern GOMx have decreased [25]. Overall, these trends are projected to continue at least through the end of the 21st century [26–28].

Both direct and indirect impacts of climate change can dramatically affect the distribution, behavior, and movements of marine communities, including large vertebrates such as marine mammals [29–36]. Marine mammal species with restricted geographical distributions and habitat tolerances will have limited opportunities to adapt to the changing conditions of their environment [30]. Furthermore, marine mammals rely on stable environments where prey availability is relatively predictable [37, 38]. Consequently, both direct and indirect impacts of climate change are imminent for marine mammals and are expected to continue [30, 39–41]. Indeed, some marine mammal populations (e.g., ice-associated seals, subarctic cetaceans) have already shown climate-related shifts in distribution [42–45]. Although predicting climate-driven changes in marine mammal distribution, phenology, and abundance is challenging [46–48], new modeling tools and approaches have improved our predictive capabilities [47, 49–52].

A variety of approaches have been used to characterize climate impacts on living marine resources. These include habitat suitability models (e.g., [53]), scenario planning [54, 55], and climate vulnerability assessments (CVAs) [56, 57]. CVAs identify the factors contributing to species' vulnerability to climate change and rank or categorize species that may be most vulnerable using a rapid but generalized approach that typically combines exposure, sensitivity, and adaptive capacity [57–60]. There have been numerous CVA studies of terrestrial species dating back to the 1990s (e.g., [61, 62]). However, CVAs are less common for marine ecosystems [56, 63–65], with marine fisheries and marine and coastal habitat receiving the most attention to date (e.g., [66–71]). In recent years, CVAs, and elements thereof, have been applied to marine mammals [72–76].

Trait-based CVAs often rely on qualitative assessments of species' biological or ecological traits that have been linked with climate responses [57, 77] and may suffer where those linkages are not well established [57, 78]. For ecosystems where linkages are well established [79], trait-based CVAs can effectively incorporate expert elicitation to address areas with data deficiencies [69, 70]. Although trait-based CVAs provide less resolution than some quantitative methods, they can play an important role in identifying and planning for climate change impacts to species due to their rapid and adaptable approach [59, 80].

Given the challenges presented by changing climate conditions, conservation and protection of species and populations can no longer be viewed through traditional lenses, and climate change must be considered to adequately manage species [81, 82]. To aid management and conservation strategies, an improved understanding of marine mammal responses to altered climate states is needed at the management unit, or "stock" level. In the United States, the National Oceanic and Atmospheric Administration's (NOAA) National Marine Fisheries Service (NOAA Fisheries) and the United States Fish and Wildlife Service (USFWS) have a mandate to protect and recover marine mammal species under the Endangered Species Act (ESA) and Marine Mammal Protection Act (MMPA). The MMPA defines a stock as "a group of marine mammals of the same species in a common spatial arrangement that interbreed when mature" (MMPA, 16 USC, 1361 et seq.). Stocks of marine mammals are defined for the purpose of stock assessments, which can be used to identify and potentially mitigate the effects of anthropogenic and natural stressors. Here, we conducted a trait-based CVA of United States marine mammal stocks in WNA, GOMx, and Caribbean waters, to provide the first assessment of potential, climate-associated threats to marine mammals in these waters. We present a ranked list of stocks by climate vulnerability score, an assessment of the confidence of those scores, and the primary ecological and environmental drivers of climate vulnerability.

## Methods

### Overview

We followed the approach outlined by the NOAA Fisheries Marine Mammal Climate Vulnerability Assessment (MMCVA) [83], based on the NOAA Fisheries Marine Fish and Invertebrate Climate Vulnerability Assessment (FCVA) [70, 84]. The MMCVA is a modified Delphi approach [85–87] that uses expert elicitation through iterative rounds of scoring and discussion. Experts scored two separate components for each marine mammal stock: (1) degree of exposure to climate change; and (2) sensitivity and capacity to adapt to climate change. These two component scores were then combined to calculate a relative climate vulnerability score.

We convened an expert workshop in July 2015, with representatives from NOAA, other governmental agencies, non-governmental organizations (NGOs), and academia to inform the selection of relevant climate exposure factors and climate sensitivity attributes [83]. We used existing CVA frameworks and approaches as models (e.g., [69, 70, 73, 88]) and used prior syntheses of climate impacts on marine mammals (e.g., [31, 89, 90]) to establish elements of the two components of the assessment.

### Exposure component

Climate exposure was defined as the magnitude of environmental change a marine mammal stock is expected to experience within its current geographic distribution. Climate exposure was scored using 16 abiotic exposure factors that are likely to affect marine mammals, their prey, and/or their habitat (Table 1) [83]. Seven of these factors (SST, air temperature, precipitation, salinity, ocean pH, sea ice cover, and dissolved oxygen) were modeled in the NOAA Climate Change Web Portal [91] using the ensemble of models from the Coupled Model Intercomparison Project phase 5 (CMIP5) [92] with representative concentration pathway (RCP) 8.5 [93, 94], using the historical period of 1956–2005 and future period of 2006–2055. RCP 8.5 represents a "business-as-usual" emissions scenario that assumes the fewest greenhouse gas mitigation measures will be implemented [94] and is used by NOAA Fisheries when considering the treatment of climate change in ESA decisions [95]. Through a series of scoping conversations during the method development, a mid-century future time frame was determined to be near-term enough to be useful to managers for statutory and management activities (e.g., recovery planning, critical habitat designation, biological opinions) [83]. We scored the seven factors from the NOAA Climate Change Web Portal using two metrics: projected change in mean and variability of the climate factor. The change in mean condition of a factor was represented by a projected future standard anomaly, which compares projected future conditions to historical conditions by subtracting the historical mean from the projected future mean and then dividing the difference by the historical standard deviation. The change in variability of a factor was represented by an F-ratio, calculated as future variance divided by historical variance. We scored circulation qualitatively by evaluating the types of circulation (e.g., wind-driven, tidal) with which each stock interacts and how those types of circulation may change. Interaction with large boundary currents or astronomically driven circulation represented less exposure to climate change while interaction with wind- or weather-driven circulation (e.g., upwelling) represented greater exposure to climate change (Table 1) [70, 84]. Sea level rise was scored using projected relative sea level change at 2060 [96], which was the nearest time frame to the 2006–2055 time frame used for the other exposure factors. Scoring the sea level rise exposure factor also included a qualitative element such that stocks in deep water scored as low exposure (Table 1). Circulation and sea level rise did not include a variability metric.

**Table 1. Exposure factors and scoring criteria used in assessing climate vulnerabilities of 108 marine mammal stocks in the western North Atlantic, Gulf of Mexico, and Caribbean Sea.**

| Exposure Factor | Metric | Low | Moderate | High | Very High |
|---|---|---|---|---|---|
| Sea Surface Temperature Air Temperature Precipitation Salinity Ocean Acidification Sea Ice Cover Dissolved Oxygen | Change in mean | $\|x\| < 0.5$ std dev | 0.5 std dev $\leq \|x\| <$ 1.5 std dev | 1.5 std dev $\leq \|x\| <$ 2.0 std dev | $\|x\| \geq 2.0$ std dev |
| Sea Surface Temperature Air Temperature Precipitation Salinity Ocean Acidification Sea Ice Cover Dissolved Oxygen | Change in variability | F ratio <1.15 | 1.15 $\leq$ F ratio < 1.54 | 1.54 $\leq$ F ratio < 1.78 | F ratio $\geq$ 1.78 |
| Circulation | Qualitative | Stock distribution overlaps almost exclusively with large boundary currents or tidal currents | Majority of stock distribution overlaps with large boundary currents or tidal currents. Stock may also interact with mesoscale features such as fronts or eddies | Majority of stock distribution overlaps with currents that are expected to have a high magnitude of change such as estuarine circulation, nearshore density currents, and/or wind driven currents. Stock may also interact with mesoscale features such as fronts or eddies | Stock distribution overlaps almost exclusively with currents that are expected to have a high magnitude of change such as estuarine circulation, nearshore density currents, and/or wind driven currents |
| Sea Level Rise | Semi-qualitative | Stock is found generally in deeper water beyond the continental shelf | Stock is generally coastal or found in continental shelf waters | Stock relies on wetland, seagrass, beach, or estuary habitat for one or more life stage and the change in regional sea level within their range is expected to increase less than 7 mm yr-1 by 2050 | Stock relies on wetland, seagrass, beach, or estuary habitat for one or more life stage and regional sea level within their range is expected to increase greater than or equal to 7 mm yr-1 by 2050 |

## Sensitivity component

Sensitivity was defined as the ability of a stock to tolerate climate-driven changes in environmental conditions. We included elements of adaptive capacity within the sensitivity component. Adaptive capacity was defined as the ability to modify intrinsic characteristics through behavioral or evolutionary processes to cope with climate-driven changes in environmental conditions [59]. Tolerance of a condition and adaptation to a condition exist along a spectrum of possible responses to that condition, resulting in attributes that could be categorized as relating to sensitivity or relating to adaptive capacity with simple changes in wording [70, 97]. For example, a stock with a generalist diet could be considered to have low sensitivity and be highly adaptive to climate-driven changes in its prey. Therefore, we combined sensitivity and adaptive capacity into a single component, hereafter referred to as the "sensitivity component" and the attributes within it referred to as "sensitivity attributes." The sensitivity component was scored using 11 life history traits and ecological variables related to climate change (Table 2) [83].

**Table 2. Sensitivity attributes and scoring criteria used to assess climate vulnerabilities of 108 marine mammal stocks in the western North Atlantic, Gulf of Mexico, and Caribbean Sea.**

| Sensitivity Attribute | Sensitivity Score | | | |
|---|---|---|---|---|
| | **Low** | **Moderate** | **High** | **Very High** |
| *Prey/Diet Specificity* | Generalist; feeds on a wide range of prey types and sizes | Generalist; feeds on a limited number of prey types or sizes, but a wide variety of species within those types | Specialist; exhibits strong preference for one prey type for the majority of its caloric intake, but is capable of switching prey types | Specialist; reliant on one prey type, often a single genus or family, for the majority of its caloric intake, and is unable to switch to other prey types |
| *Habitat Specificity* | Stock exclusively utilizes physical features resilient to climate conditions | Stock utilizes a variety of features, but is not reliant on physical features vulnerable to climate conditions and/ or biogenic habitat for specific life stages | Stock relies on biogenic habitat or physical features vulnerable to climate conditions for one life stage or event | Stock relies on biogenic habitat or physical features vulnerable to climate conditions for multiple life stages or events, or for any one particularly critical life stage or event |
| *Site Fidelity* | Individuals display no site fidelity | Individuals display a low degree of site fidelity (i.e., archipelagos or coastlines of a general region) | Individuals display a high degree of site fidelity (i.e., specific islands or bays) for either foraging or breeding | Individuals display a high degree of site fidelity (i.e., specific islands or bays) for both foraging and breeding |
| *Lifetime Reproductive Potential* | High reproductive output based on reproductive lifespan and reproductive interval | High-moderate reproductive output based on reproductive lifespan and reproductive interval | Low-moderate reproductive output based on reproductive lifespan and reproductive interval | Low reproductive output based on reproductive lifespan and reproductive interval |
| *Generation Time* | < 10 years | 10 years ≤ x < 20 years | 20 years ≤ x < 30 years | ≥ 30 years |
| *Reproductive Plasticity* | High reproductive plasticity (e.g., long breeding season, broad breeding range, few breeding habitat requirements) | High-moderate reproductive plasticity | Low-moderate reproductive plasticity | Low reproductive plasticity (e.g., short breeding season, narrow breeding range, strict breeding habitat requirements) |
| *Migration* | Annual migration; multiple migratory pathways | Annual migration; single migratory pathway | Seasonal migration | No migration; local movement only |
| *Home Range* | Individuals' home ranges are broad (e.g., include much of an ocean basin) | Individuals' home ranges are moderate to large (e.g., spend the majority of time along coasts, within continental shelf waters, or along the continental slope, but may utilize deeper waters) | Individuals typically remain in bays or archipelagos and seldom travel farther but could if needed | Individuals' home ranges are relatively small (e.g., confined to bays or archipelagos) and are limited from traveling farther by a combination of geographic features, physical capabilities, and behaviors |
| *Stock Abundance* | Stock comprises > 10,000 individuals | Stock comprises 1,001–10,000 individuals | Stock comprises 101–1,000 individuals | Stock comprises < 100 individuals |
| *Stock Abundance Trend* | Increasing abundance trend over past 10-year period | Stable abundance trend over past 10-year period | Declining abundance trend over past 10-year period | Rapidly declining abundance trend over past 10-year period |
| *Cumulative Stressors* | Stock currently experiences 1 or fewer additional stressors | Stock currently experiences 2 or 3 additional stressors | Stock currently experiences 4 or 5 additional stressors | Stock currently experiences greater than 5 additional stressors or has one additional stressor that accounts for more than half of annual mortality |

## Data quality

We scored the quality of available data for each factor and attribute by asking each expert to assign a data quality score as described in Table 3 [70, 84]. For exposure factors, data from the NOAA Climate Change Web Portal were considered to be high quality and data quality scores referred to the certainty of distribution for each stock relative to that exposure factor. For sensitivity attributes, these data quality scores referred to the data provided to and by the experts in the stock narratives.

**Table 3. Data quality score definitions used in the climate vulnerability assessment of 108 marine mammal stocks in the western North Atlantic, Gulf of Mexico, and Caribbean Sea.**

| Data Quality Score | Criteria |
|---|---|
| 3 | Observed, modeled, or measured data support tally placement. |
| 2 | Observed, modeled, or measured data from similar stocks or species support the tally placement. Dated or conflicting information complicates the ability to place tallies. |
| 1 | Expert's knowledge of and experience with the stock is the sole basis for tally placement. |
| 0 | No information is available to support tally placement and the expert's familiarity with the stock is insufficient to provide expert judgment. |

## Scope

We conducted this assessment using 108 cetacean and pinniped stocks managed by NOAA Fisheries in the WNA, GOMx, and Caribbean Sea (S1) [98]. In some instances, stocks or species within the same genus were combined into a single stock group (see S1). For example, the pygmy sperm whale (*Kogia breviceps*) and dwarf sperm whale (*Kogia sima*) were combined into a single *Kogia* sp. group due to the difficulty in visually differentiating these species and the frequency in which they are discussed together in the literature (e.g., [99]). In addition, each stock or stock group (hereafter referred to as a "stock") was classified into a taxonomic group (delphinid, mysticete, other odontocete, pinniped, ziphiid) to allow for generalized assessment. For stocks whose individual ranges extended beyond the United States Exclusive Economic Zone (EEZ), we included the full ranges within the scope of the assessment. For example, North Atlantic right whales (*Eubalaena glacialis*) travel into Canadian waters and their occurrence in those waters was included here. West Indian manatees (*Trichechus manatus*), which are managed by the USFWS, were not included in this assessment.

## Participants

In total, 41 marine mammal experts scored stocks in this assessment. These subject matter experts were broadly familiar with their assigned stocks and species through field or other research experience. While expertise in any given stock was valuable, the ability of experts to score multiple stocks allowed for comparisons across stocks. Expert scorers included staff from NOAA and other federal government agencies, NGOs, and academia. All expert scorers are included here as co-authors.

## Assessment resources

**Background documents.** Prior to the assessment, we assembled information about each stock's ecology and life history related to the sensitivity attributes (Table 2), the stock's geographic distribution, and any peer-reviewed publications about the stock relating to climate change. We organized this information into stock narratives, similar to other CVAs (e.g., [66, 69, 70]) and continued to update the narratives as new publications became available during the assessment (see S2). For understudied stocks, we included ecology and life history information from related stocks or species. For example, if reproductive information about the common bottlenose dolphin (*Tursiops truncatus*) Caloosahatchee River Stock was missing, information from the nearby common bottlenose dolphin Sarasota Bay/Little Sarasota Bay Stock was included as a reference point.

**Exposure maps.** To facilitate scoring exposure factors, we created a set of exposure maps for each stock. Climate projection datasets were queried and downloaded using the settings described in Table 4. We used ArcGIS 10.5.1 (Esri, Redlands, CA, USA) to create maps for

**Table 4. Values used in the NOAA climate change web portal to generate climate exposure maps for 108 marine mammal stocks in the western North Atlantic, Gulf of Mexico, and Caribbean Sea.**

| Field | Value |
|---|---|
| Experiment | RCP 8.5 |
| Model | Average of All Models |
| Variable | [based on climate exposure factor] |
| Statistic | Standard Anomaly (average historical) |
| Season | Entire year OR specific season for highly migratory stocks |
| 21st Century Period | 2006–2055 |
| Region | Scaled to fit entire stock distribution |

each exposure factor by reclassifying projected values according to the bin scoring criteria (Table 1). Sea level rise exposure maps were generated using the nearest geographic tide gauge point estimate of regional sea level rise in 2060 under the Intermediate-Low (0.5m global sea level rise by 2100) scenario [96], which falls within the likely range of global sea level rise of 0.45–0.82 m by 2100 relative to 1986–2005 projected under RCP 8.5 [100].

Sightings data obtained from the Ocean Biodiversity Information System Spatial Ecological Analysis of Megavertebrate Populations (OBIS-SEAMAP) [101] were superimposed onto climate exposure maps to provide a range context for each stock. Stock boundaries were also superimposed where available. Experts were supplemented with NOAA marine mammal stock assessment reports (e.g., [98]) and density plots from other published studies (e.g., [102, 103]).

## Scoring approach

Following the model of the FCVA [70, 84], each exposure factor and sensitivity attribute was scored individually by a set of experts for a given stock. For each exposure factor and sensitivity attribute, experts were allotted five points that could be applied across four scoring bins (Bin 1 = low, Bin 2 = moderate, Bin 3 = high, Bin 4 = very high). Scoring bins were delineated by criteria specific to that factor or attribute (Tables 1 and 2).

Experts first individually scored exposure factors, sensitivity attributes, and the data quality of assigned stocks during a preliminary scoring round (December 2017 –February 2018). In March 2018, we held a full group debrief webinar to discuss major trends in scoring. We clarified questions about the scoring criteria and provided guidance to ensure a consistent scoring process. For stocks that had major scoring differences between scorers, we held stock-specific conference calls to allow experts to discuss scores. Experts individually revised and updated their scores during the final scoring round (April 2018 –September 2018). We aimed for each stock to be scored by at least three experts, and accomplished this for all but one stock (the Northern North Carolina Estuarine System Stock of common bottlenose dolphins had only two scorers). Of the 108 marine mammal stocks and stock groups, 56% (n = 60) of stocks had three scorers, 34% (n = 37) of stocks had four scorers, 8% (n = 9) of stocks had five scorers, and 1% (n = 1) had six scorers. The number of scorers varied by stock as a result of variations in the number of researchers available with expertise for a given stock.

Experts compared the geographic distribution of each stock (based on range maps, sighting data, and density plots from the published literature) to the projected exposure level for each individual factor and scored each exposure factor by placing five tallies across the four bins according to the magnitude of exposure projected across the entirety of the stock's current distribution. For example, if the entire stock distribution had a magnitude of exposure that matched the criteria for "Bin 4," all five tallies were placed in "Bin 4." If part of the stock distribution had a magnitude of exposure that matched the criteria for "Bin 4" and part that

matched the criteria for "Bin 3," experts placed tallies according to the proportion of the distribution that matched each bin. Circulation was scored using expert judgement based on the interaction of the stock with various types of currents. Sea level rise was scored using a combination of expert judgement based on the literature about projected impacts and overlap of the stock's distribution with projected sea level rise.

Similarly, experts used their knowledge and experience with the stock or species combined with the stock narratives to place their five tallies into the four sensitivity bins. For example, if all supporting evidence matched the criteria in "Bin 4," the experts placed all five tallies in "Bin 4." If evidence for a stock ranged across several bins, experts could spread their tallies across multiple bins based on the supporting evidence and their expert judgment.

## Analyses

To calculate the overall climate vulnerability scores, we followed a process similar to the FCVA [70, 84]. First, we calculated weighted mean scores for each exposure factor and sensitivity attribute using the tallies from all experts for each stock/factor/attribute combination and the following equation:

$$Factor\ or\ Attribute\ Weighted\ Mean\ Score\ = \frac{(B_1 * 1)\ +\ (B_2 * 2)\ +\ (B_3 * 3)\ +\ (B_4 * 4)}{B_1\ +\ B_2\ +\ B_3\ +\ B_4},$$

where $B_1$–$B_4$ are the number of tallies within each bin and the multipliers in the numerator are the weighting value for each bin. Exposure and sensitivity component scores for a stock were determined from the factor and attribute weighted mean scores using the logic model from the FCVA (Table 5) [70, 84].

The overall vulnerability of a stock was determined by combining exposure component scores and sensitivity component scores to generate a vulnerability rank and place the stock into a vulnerability category. The higher scores correlate to greater stock vulnerability. Stocks were placed into vulnerability categories using the exposure component score and sensitivity component score cross-referenced with a vulnerability matrix derived from the FCVA [70, 84]. We combined exposure and sensitivity component scores to calculate relative overall climate vulnerability scores. Scores were combined into relative vulnerability categories of low, moderate, high, and very high.

We conducted a bootstrap analysis using R (version 3.6.2) [104] to estimate the certainty of the climate vulnerability scores. In this context, certainty is a measure of the consistency of the original score with the bootstrap score. The bootstrap score provides an estimate of the vulnerability score sensitivity given the variance in the scoring distribution. For each stock, we sampled with replacement the tallies of all experts for each exposure factor and sensitivity attribute. We recalculated the sensitivity score, exposure score, and vulnerability score for each of 10,000 iterations. We reported score certainty as the proportion of those 10,000 iterations that scored in each bin.

**Table 5. Logic model used to convert attribute and factor scores to component scores.**

| Component Score | Criteria |
|---|---|
| Very High (4) | 3 or more attribute or factor mean scores $\geq$ 3.5 |
| High (3) | 2 or more attribute or factor mean scores $\geq$ 3.0, but does not meet threshold for "Very High" |
| Moderate (2) | 2 or more attribute or factor mean scores $\geq$ 2.5, but does not meet threshold for "High" or "Very High" |
| Low (1) | Less than 2 attribute or factor mean scores $\geq$ 2.5 |

**Distribution, abundance, and phenology responses.** A stock's response to climate change may manifest in a variety of ways including: 1) shifts in distribution resulting in climate-driven changes in geographic ranges, including range expansion, contraction, or alteration; 2) fluctuations in abundance resulting in declines or increases; and/or 3) variations in phenology resulting in seasonal shifts (either earlier and/or later in the year) or temporal changes (prolonged and/or shortened) of life history events such as breeding or migration. We considered whether each sensitivity attribute could be expected to drive potential responses in a stock's abundance, geographic distribution, and/or phenology (Table 6). Some attributes influenced all three response categories, while other attributes only influenced one or two response categories.

Mean sensitivity attribute scores were placed with the response categories (abundance, distribution, phenology) identified as relevant to that attribute (see Appendix A in Lettrich et al. [83]). For example, if a given attribute was determined to have influence over all three response categories, then the mean attribute score applied to each response category. If a given attribute was determined to have influence over only abundance, the mean attribute score was applied to abundance, but not to the distribution or phenology for that attribute. The three response categories remained independent of one another and were supplemental to the mean sensitivity attribute score.

To estimate the potential for shifts in distribution, declines in abundance, and variations in phenology, we calculated response scores based on a subset of sensitivity attributes for each response type. We characterized each response type as directly or inversely related to the attribute score (Table 6). Some attributes were viewed as inversely related due to the nature of the response category. Full descriptions are found in Appendix A of Lettrich et al. [83].

**Determining attribute importance and expert effect: Leave-one-out analysis.** We recalculated sensitivity and vulnerability scores for each stock by sequentially omitting each sensitivity attribute. We reported the influence of each sensitivity attribute as the change in sensitivity score and vulnerability score by omitting that sensitivity attribute. We conducted a similar analysis for the effect each expert had on vulnerability scores by sequentially omitting the scores of each expert and recalculating each stocks' vulnerability score.

## Results

### Overall vulnerability

Of the 108 marine mammal stocks and stock groups, 44% (n = 47) scored as having very high vulnerability to climate change, 29% (n = 31) scored high, 20% (n = 22) scored moderate, and 7% (n = 8) scored low (Fig 1 and S2 Table). The majority of stocks (72%, n = 78) scored as having very high exposure (Fig 1 and S2 Table) while sensitivity scores showed a more even spread: 33% (n = 36) of stocks scored as having very high sensitivity, 18% (n = 19) scored high, 34% (n = 37) scored moderate, and 15% (n = 16) scored low (Fig 1 and S2 Table).

The bootstrap analysis showed that 43% (n = 46) of stocks scored $\geq$ 90% certainty, 27% (n = 29) had certainty scores 66–89%, and 31% (n = 33) had certainty scores $\leq$ 66%. The original vulnerability score matched the score with the greatest proportion of iterations in 90.7% (n = 98) of stocks, while 4.6% (n = 5) of stocks increased in score and 4.6% (n = 5) decreased in score in the bootstrap analysis. Sensitivity scores changed more than exposure scores in the bootstrap analysis

### Distribution, abundance and phenology

We used nine of the eleven sensitivity attributes to estimate the possibility that a stock would experience a shift in distribution (Table 6), and these results identified 12% (n = 13) of stocks

**Table 6. Response variable ordination used in climate vulnerability assessment of 108 stocks of marine mammals from the western North Atlantic, Gulf of Mexico, and Caribbean Sea.**

| Sensitivity Attribute | Distribution | Abundance | Phenology |
|---|---|---|---|
| Prey/Diet Specificity | Direct* | Direct | Direct |
| Habitat Specificity | Direct | Direct | Direct |
| Site Fidelity | Inverse | Direct | N/A |
| Lifetime Reproductive Potential | N/A | Direct | N/A |
| Generation Time | N/A | Direct | N/A |
| Reproductive Plasticity | Inverse | Direct | Inverse |
| Migration | Inverse | N/A | Inverse |
| Home Range | Direct | Direct | N/A |
| Stock Abundance | Direct | Direct | Direct |
| Stock Abundance Trend | Direct | Direct | Direct |
| Cumulative Stressors | Direct | Direct | Direct |

*"Direct" and "inverse" refer to the ordination relative to the scoring criteria in Table 2.

as having a very high possibility of experiencing shifts in distribution, 69% (n = 75) as high, 18% (n = 19) as moderate, and 1% (n = 1) as low (S3). We used ten sensitivity attributes to estimate the potential for a stock to experience a decline in abundance (Table 6). These results identified 15% (n = 16) of stocks as having a very high possibility of experiencing declines in abundance, 31% (n = 33) as high, 30% (n = 32) as moderate, and 25% (n = 27) as low (S3). We used seven of the eleven sensitivity attributes to estimate the possibility of a stock to experience a shift in phenology (Table 6). These results identified 9% (n = 10) of stocks as having a very high possibility of experiencing shifts in phenology, 43% (n = 46) as high, 34% (n = 37) as moderate, and 14% (n = 15) as low (S3).

## Regional differences

Sensitivity, exposure, and vulnerability scores varied spatially across the WNA, GOMx, and Caribbean regions. For exposure, stocks in the WNA (n = 48) scored very high (52%, n = 25), high (40%, n = 19), and moderate (8%, n = 4); for sensitivity, stocks in the WNA spanned the range of sensitivity, with the most stocks (42%, n = 20) scoring as moderate; and they spanned the range of vulnerability, with a relatively even spread across very high (25%, n = 12), high (27%, n = 13), moderate (31%, n = 15), and low (17%, n = 8) vulnerability. Stocks in the GOMx (n = 54) scored high or very high for exposure, with the vast majority (89%, n = 48) very high; for sensitivity, stocks spanned the range of sensitivity, with nearly half (46%, n = 25) scoring very high; and for vulnerability, stocks scored very high (65%, n = 35), high (22%, n = 12), and moderate (13%, n = 7). Within the Caribbean Sea, stocks (n = 6) scored very high (83%, n = 5) and high (17%, n = 1) for exposure; high (17%, n = 1) and moderate (83%, n = 5) for sensitivity; and all scored high for vulnerability.

## Taxonomic group differences

Taxonomic groupings showed differences in exposure, sensitivity, and vulnerability (Fig 2). The pinniped group (n = 4) was the only group that did not include very high exposure. The mysticete group (n = 7) included stocks with only high (29%, n = 2) and very high (71%, n = 5) exposure but spanned the full range of sensitivity. No mysticete stocks scored as low vulnerability and the majority (57%, n = 4) scored as high vulnerability. The ziphiid group (n = 8) scored only as high (13%, n = 1) and moderate (88%, n = 7) sensitivity. Stocks in the "other

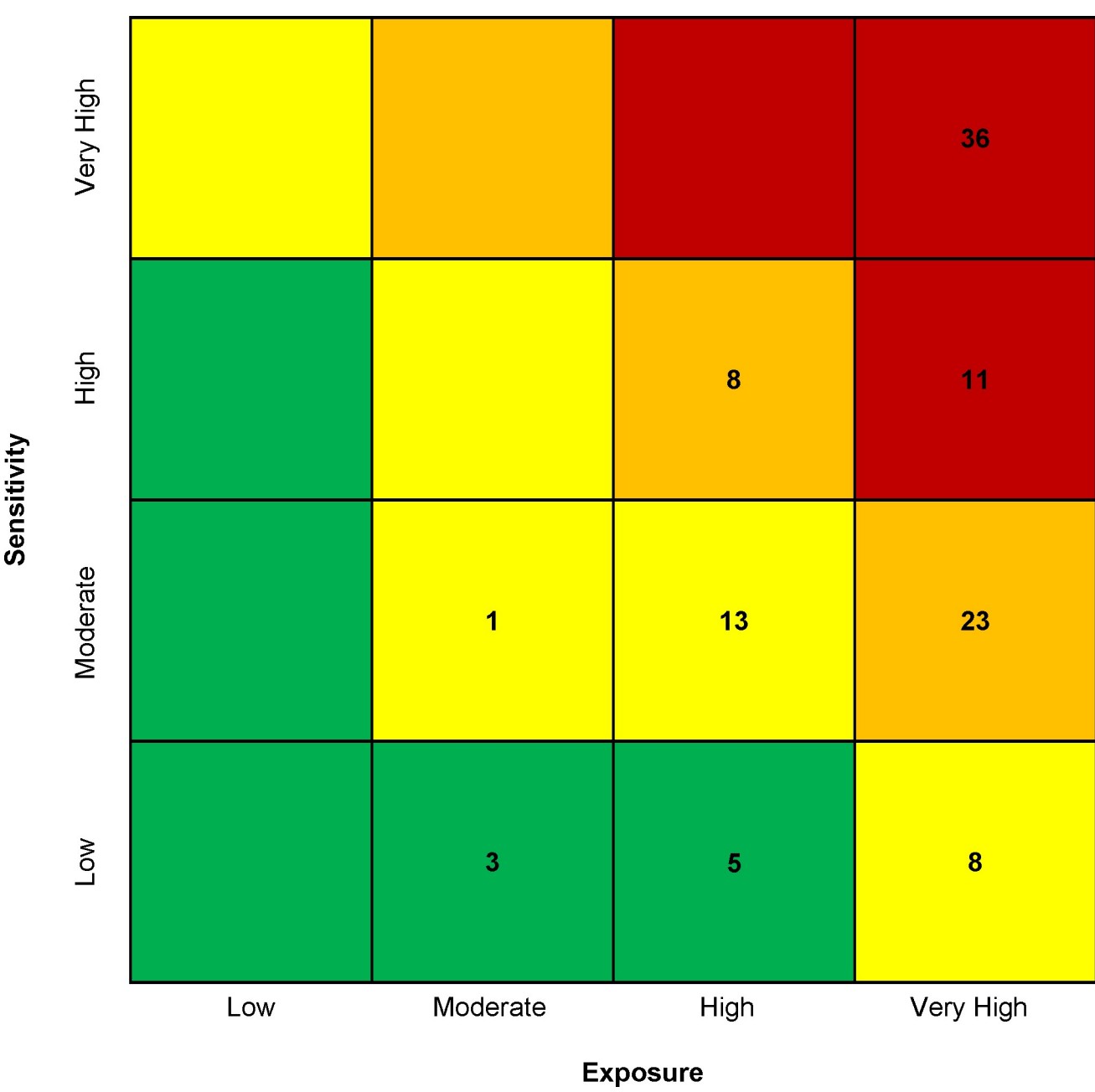

**Fig 1. Climate vulnerability matrix showing the number of marine mammal stocks for each sensitivity-exposure score combination.** Climate vulnerability is represented by cell color (green = low vulnerability, yellow = moderate vulnerability, orange = high vulnerability, red = very high vulnerability). Numbers indicate the number of stocks that scored in each sensitivity/exposure combination.

odontocete" group (*Physeter macrocephalus* and *Kogia* sp., n = 5) scored only as high (60%, n = 3) and moderate (40%, n = 2) vulnerability. Delphinid stocks (n = 84) spanned the full range of vulnerability and sensitivity, and scored moderate to very high exposure.

Common bottlenose dolphin stocks represented 47% (n = 51) of the stocks scored. We split those stocks into two groups: inshore (i.e., bay, sound, and estuary stocks; n = 42) and coastal/offshore (n = 9). The bay, sound, and estuary common bottlenose dolphin stocks scored as very high (81%, n = 34) and high (19%, n = 8) vulnerability, while coastal/offshore common

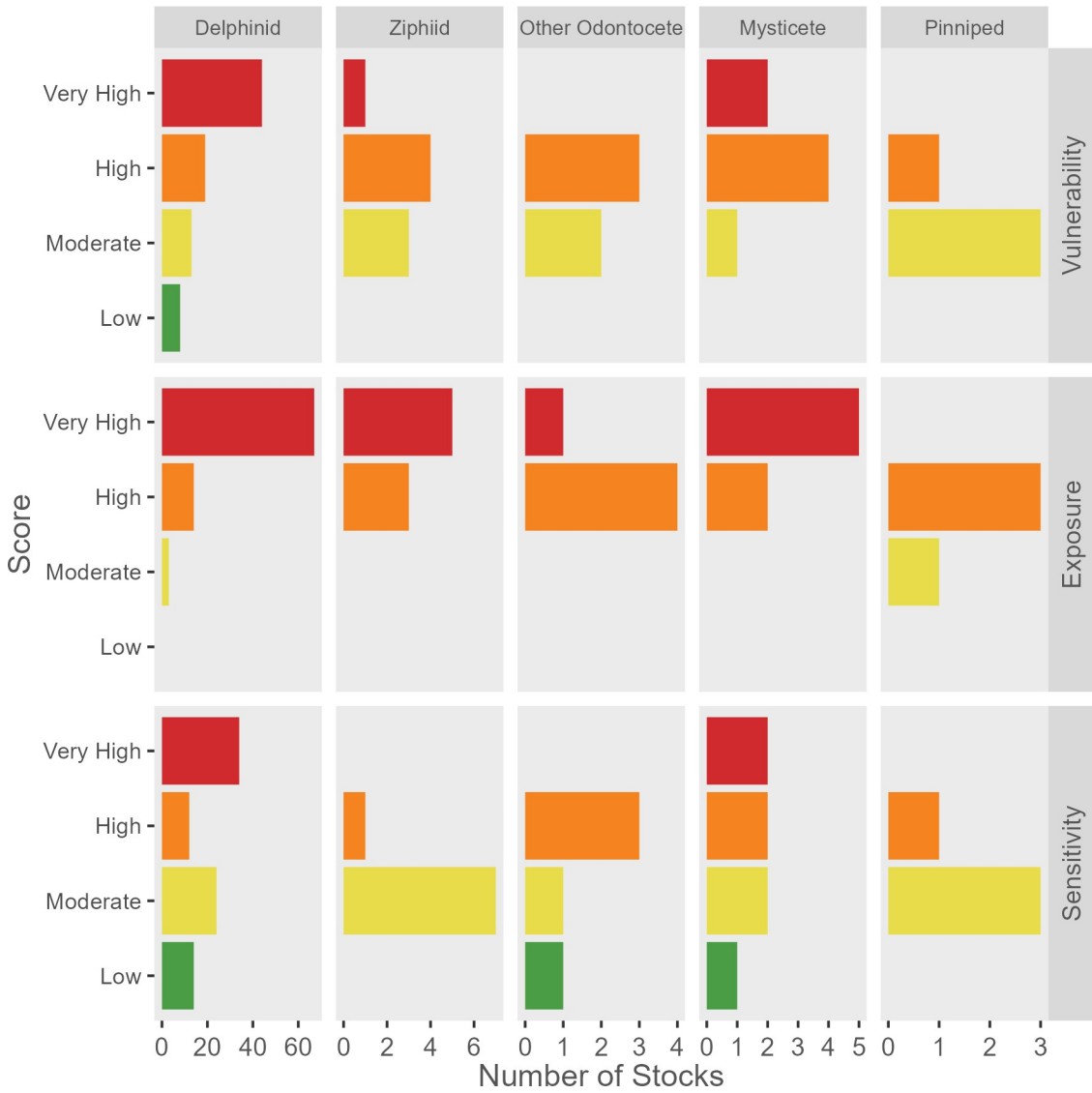

**Fig 2. Climate vulnerability, exposure, and sensitivity by taxonomic group.** Climate vulnerability, exposure, and sensitivity of U.S. marine mammal stocks in the western North Atlantic, Gulf of Mexico, and Caribbean Sea by taxonomic group. Note the different scales on the horizontal axis between taxonomic groups.

bottlenose dolphin stocks scored as high (22%, n = 2), moderate (67%, n = 6), and low (11%, n = 1) vulnerability (S3).

## Protected status differences

Of the eight stocks of species classified as "endangered" under the ESA (see S1), six scored as high vulnerability and two scored as very high vulnerability (S3). No species in the WNA, GOMx, or Caribbean Sea were listed as "threatened" under the ESA at the time of our assessment. Of the 54 stocks in this region designated as "depleted" or "strategic" under the MMPA (see S1), 7% (n = 38) scored as very high vulnerability, 26% (n = 14) scored as high vulnerability, and 4% (n = 2) scored as moderate vulnerability (S2 Table).

## Exposure factors

Air temperature, dissolved oxygen, ocean pH, and SST were the exposure factors with the highest median factor scores across all stocks. Generally, change in mean climate conditions had greater influence on exposure component scores than change in variability of climate conditions. Sea level rise showed a bi-modal distribution of scores, with few stocks scoring in Bin 3, which represented stocks that require intertidal or shallow subtidal habitat but were expected to experience a lower degree of sea level rise (Fig 3). A leave-one-out sensitivity analysis showed that ocean pH (change in mean) had the greatest ability to shift vulnerability scores (Fig 4). Without ocean pH (change in mean), 13 stocks would have shifted to a lower vulnerability score. Without dissolved oxygen (change in mean), 11 stocks would have shifted to a lower vulnerability score. No other factors changed vulnerability scores when omitted during the leave-one-out analysis.

## Sensitivity attributes

Among all stocks, migration had the highest median attribute score while reproductive plasticity and lifetime reproductive potential had the lowest median weighted average (Fig 5). Migration was the attribute with the highest median score for three of the taxonomic groups (delphinid, ziphiid, and other odontocete) while the prey and diet specificity attribute had the highest median score for the mysticete group and habitat specificity had the highest median

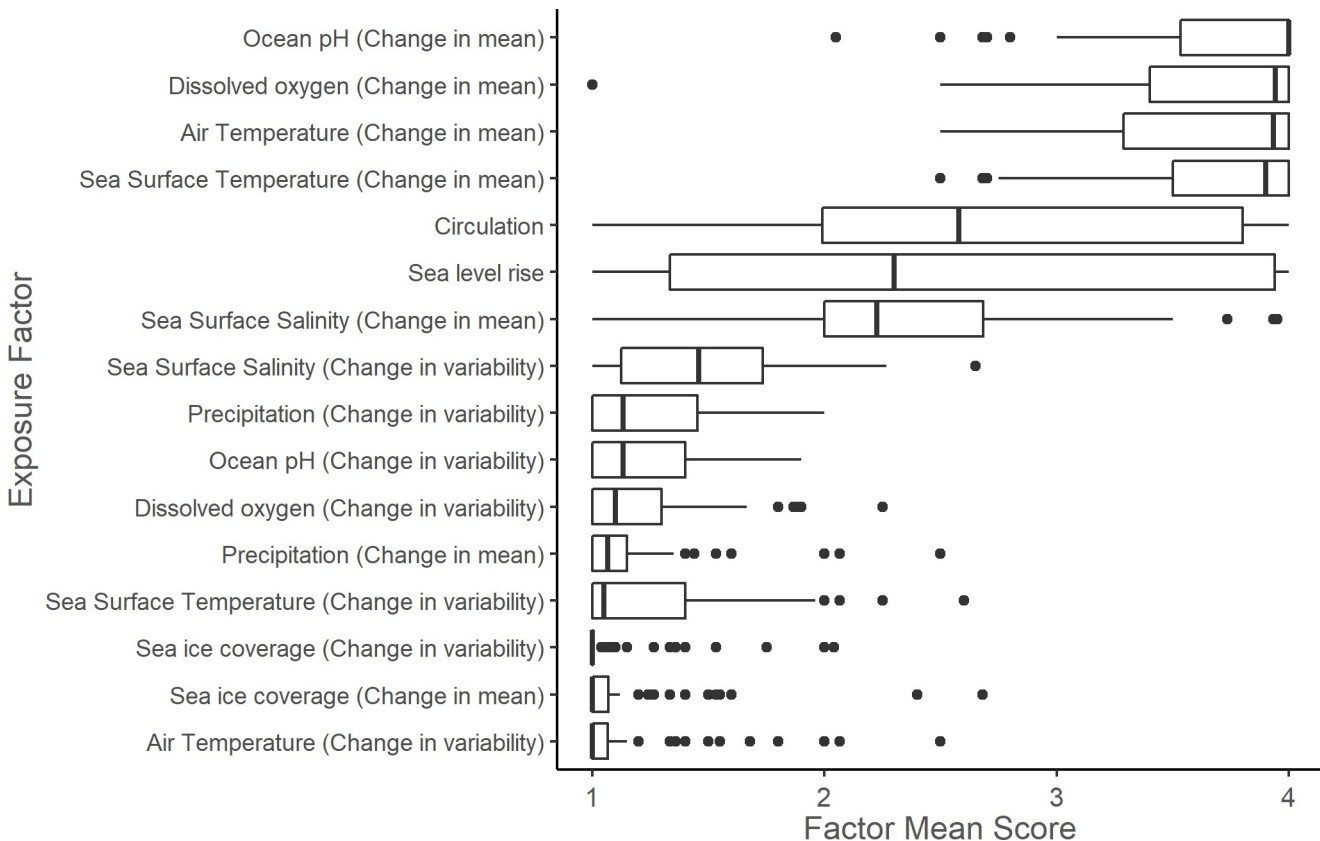

**Fig 3. Exposure factor mean scores for all scored stocks.** Exposure factor mean scores for 108 U.S. marine mammal stocks in the western North Atlantic, Gulf of Mexico, and Caribbean Sea. The vertical bar represents the median; the box is bounded by the first and third quartiles; whiskers represent 1.5 times the inter-quartile range; points represent all outlying values.

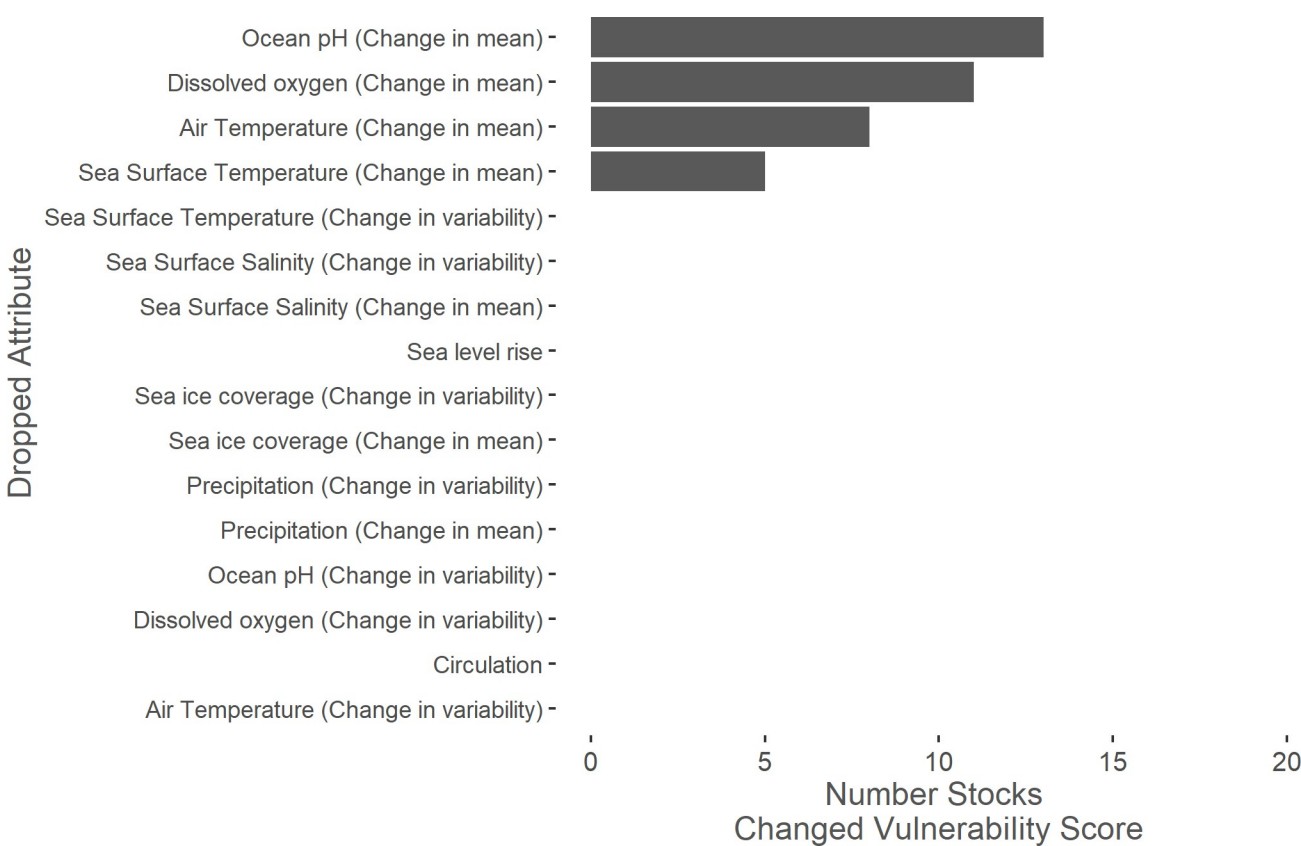

**Fig 4. Leave-one-out sensitivity analysis for exposure factors.** Leave-one-out sensitivity analysis showing how many marine mammal stocks changed climate vulnerability score when a given exposure factor was omitted. Only ocean pH (change in mean), dissolved oxygen (change in mean), air temperature (change in mean), and sea surface temperature (change in mean) changed vulnerability scores when omitted.

score for the pinniped group (see S2 File). A leave-one-out sensitivity analysis showed that migration had the greatest ability to shift vulnerability scores (Fig 6). Without the migration attribute, 16 stocks would have shifted to a lower vulnerability score (11 stocks in the delphinid group, three in the other odontocete group, and two in the ziphiid group). Without the generation length attribute (the age at which an individual has achieved half of its reproductive potential or the average age of parents of the current cohort [105, 106]), ten stocks would have shifted to a lower vulnerability score (seven stocks in the delphinid group, three in the other odontocete group).

## Expert effect

The combination of expert scoring assignments set up 383 scenarios for expert leave-one-out analysis. The effect of removing an individual expert's scores resulted in no change in vulnerability score in 77% (n = 295) of scenarios, a change in vulnerability score of one category (i.e., moving to an adjacent category) in 21% (n = 80) of cases, and a change in score of two categories (e.g., moving from low to high) in 2% (n = 8) of scenarios (see S3 File).

## Data quality

In total, 47% (n = 51) of stocks had high data quality as indicated by 80% or more of sensitivity attributes with a data quality score of two or higher, 29% (n = 31) of stocks had moderate data

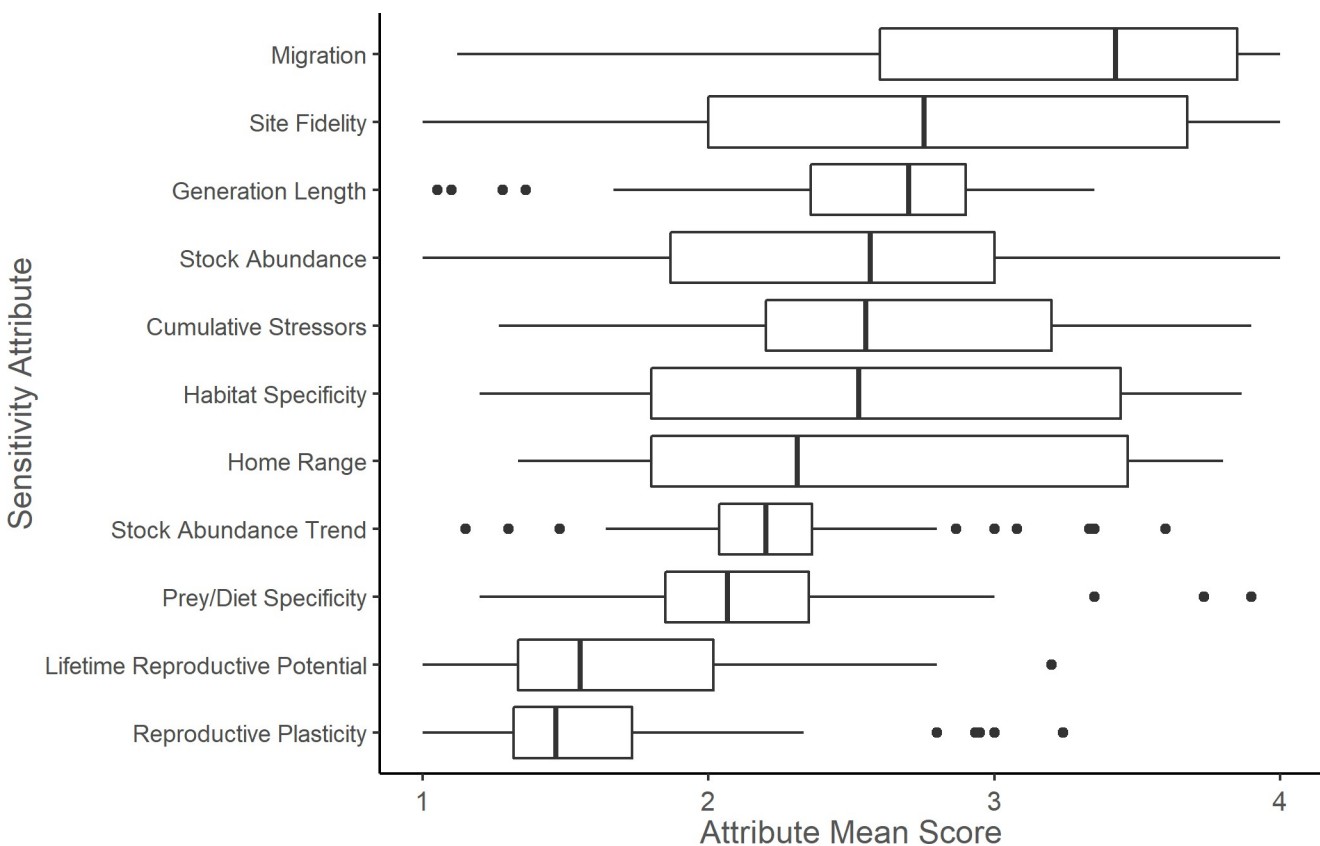

**Fig 5. Sensitivity attribute mean scores for all scored stocks.** Sensitivity attribute mean scores for 108 U.S. marine mammal stocks in the western North Atlantic, Gulf of Mexico, and Caribbean Sea. The vertical bar represents the median; the box is bounded by the first and third quartiles; whiskers represent 1.5 times the inter-quartile range; points represent all outlying values.

quality as indicated by 50%-80% of sensitivity attributes with a score of two or higher, and 14% (n = 26) of stocks had poor data quality as indicated by fewer than 50% of sensitivity attributes with a score of two or higher. Across all stocks, circulation had the lowest median exposure data quality score (Fig 7), and stock abundance trend had the lowest median sensitivity attribute data quality score (Fig 8).

## Discussion

This assessment identified that the majority (72%) of United States cetacean and pinniped stocks in the WNA, GOMx, and Caribbean Sea are considered to be highly or very highly vulnerable to climate change and variability. These regions are expected to experience a high degree of climate and environmental change by 2055 [91], which resulted in similar drivers of climate exposure and exposure scores among stocks within the region. Sensitivity attributes and the associated sensitivity scores often drove the differences in vulnerability scores between stocks.

### Exposure

Assessing exposure to climate change allows researchers and managers to better understand the extrinsic factors contributing to climate vulnerability in marine mammal stocks. Exposure scores consider the magnitude of change a stock is expected to experience within its current

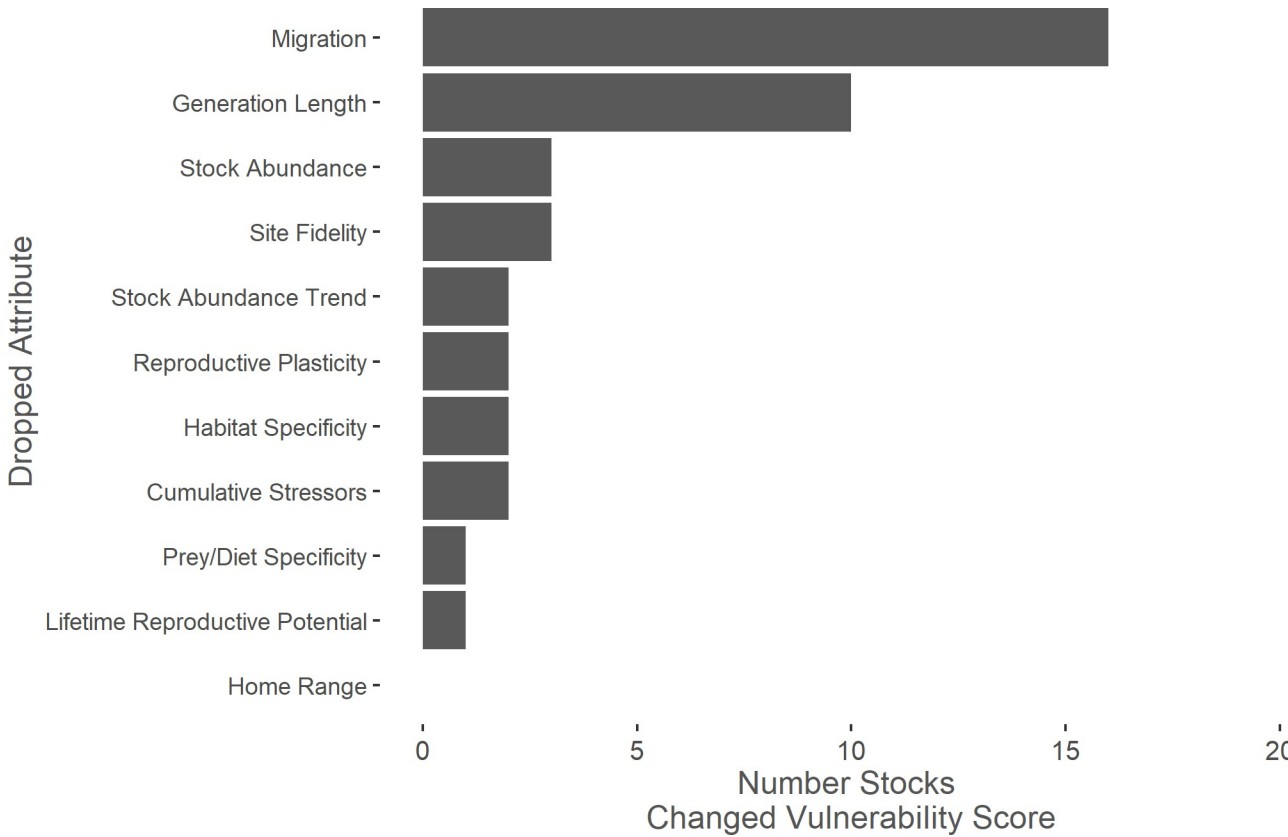

**Fig 6. Leave-one-out sensitivity analysis for sensitivity attributes.** Leave-one-out sensitivity analysis showing how many marine mammal stocks changed climate vulnerability score when a given sensitivity attribute was omitted. Home range did not change any vulnerability scores when omitted.

range relative to the recent historical variability. The most influential exposure factors included temperature, ocean pH, and dissolved oxygen (Fig 3). Temperature, both sea surface and air, was an influential factor in the exposure scores of many stocks, and past studies have shown that marine mammal distributions have been correlated with SST (e.g., [39, 107]). For example, along the New Zealand coast, some delphinid species' distributions are highly related to SST, with short-beaked common dolphins (*Delphinus delphis*) inhabiting warmer waters (>14˚C) while other species such as southern right whale dolphins (*Lissodelphis peronii*) are found inhabiting colder waters (9–16˚C) [108]. SST can also affect reproductive capacity. Elevated SST off South Georgia Island has been found to affect conception and subsequent calving rates in the eastern South American population of southern right whales (*Eubalaena australis*) [109]. SST has also been documented to affect marine mammal prey distribution, abundance, and quality (e.g., [110–115]). In terms of air temperature, its most direct effect is likely on pinnipeds while they are hauled out [116–119]. Another factor affecting pinniped species in the northeast United States is sea level rise, which can reduce or impair available habitat, particularly in the intertidal zone [71]. Air temperature has not been shown to affect cetacean thermoregulation [120, 121] and its direct effect is likely negligible on open ocean species, though precipitation and air temperature have been correlated with mortality in inshore delphinids [122]. Air temperature serves as a proxy for water temperature in estuaries, inshore waters, and shallow coastal areas that are poorly resolved by models of projected sea surface temperature [110, 111, 123, 124]. There is little evidence that ocean acidification has a direct physiological effect on marine mammals [125], but it is likely to have impacts on the distribution, abundance, phenology, and

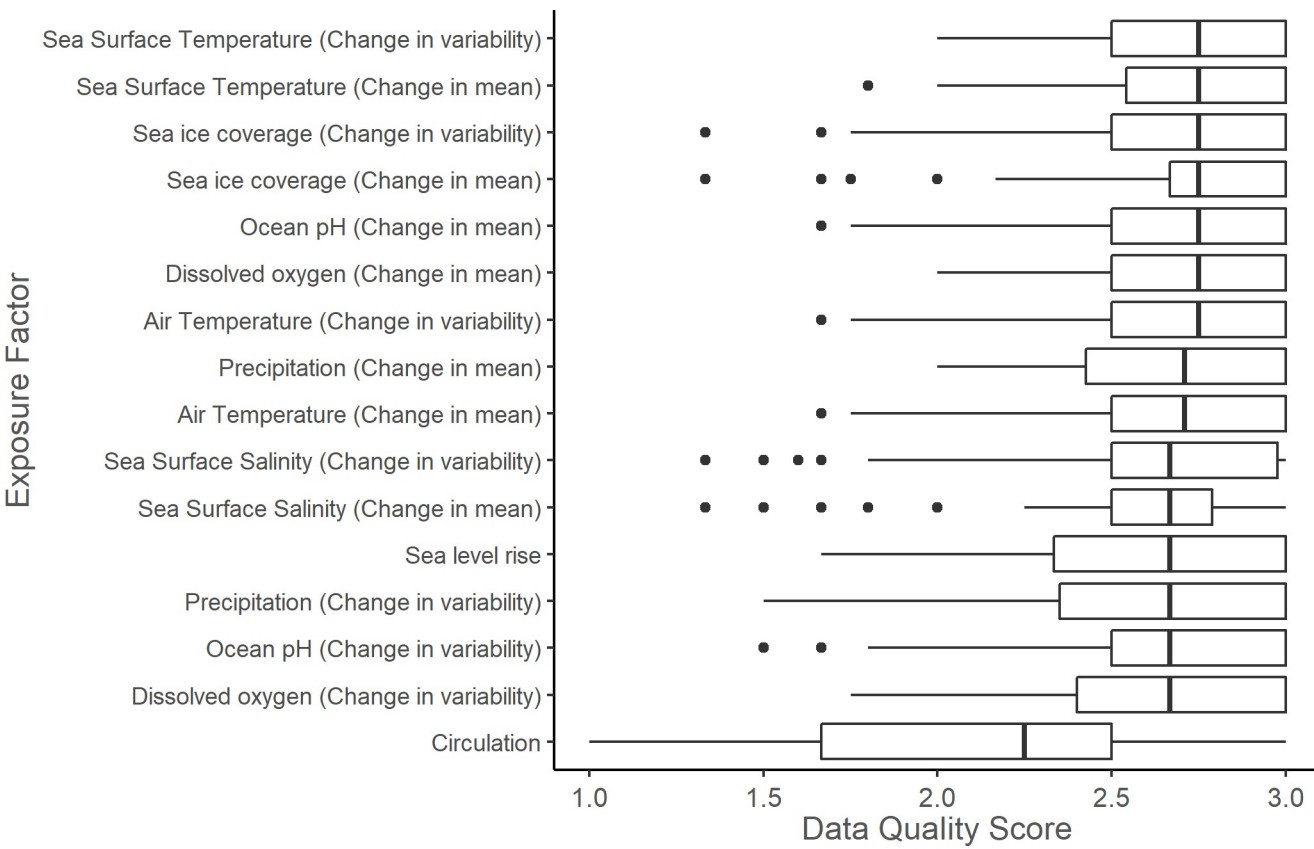

**Fig 7. Mean exposure factor data quality scores.** Mean data quality scores of climate exposure factors for 108 marine mammal stocks in the western North Atlantic, Gulf of Mexico, and Caribbean Sea. The vertical bar represents the median; the box is bounded by the first and third quartiles; whiskers represent 1.5 times the inter-quartile range; points represent all outlying values.

quality of prey species [30, 109, 126–128]. Changes in pH and temperature may have some direct impacts on marine mammals via changes in sound absorption and transmission, which could affect communication and foraging [129–132]. Although no impacts have been documented in marine mammals, changes in pH also may result in complex parasite-host relationship responses throughout the trophic web [133]. Similarly, dissolved oxygen is unlikely to have a direct effect on marine mammals, but reduced dissolved oxygen may directly impact the distribution, abundance, diversity, and richness of prey species (e.g., [134–140]).

Temperature, ocean pH, and dissolved oxygen each scored ≥3.5 in more than two-thirds of stocks. This heavy influence of a small number of exposure factors is similar to the findings of other CVAs for marine species (e.g., [70, 74, 75]). Some exposure factors were not applicable to all stocks. For instance, sea ice was not applicable to lower latitude stocks. Sea level rise and air temperature were more applicable to pinnipeds and inshore stocks than offshore stocks. Exposure scores for these 'not applicable' factors were treated as low and the use of the logic model to combine factor scores into the exposure component scores allowed these 'not applicable' factors to not affect the final scores.

Our assessment scored projected future climate change relative to recent historical variability to account for the stocks' historical experience with variable conditions. However, we acknowledge doing so may miss critical thresholds that may be crossed within the future variability. For example, a species that has historically experienced a wide range of temperatures may nonetheless be near an upper thermal threshold, which could be crossed with even a

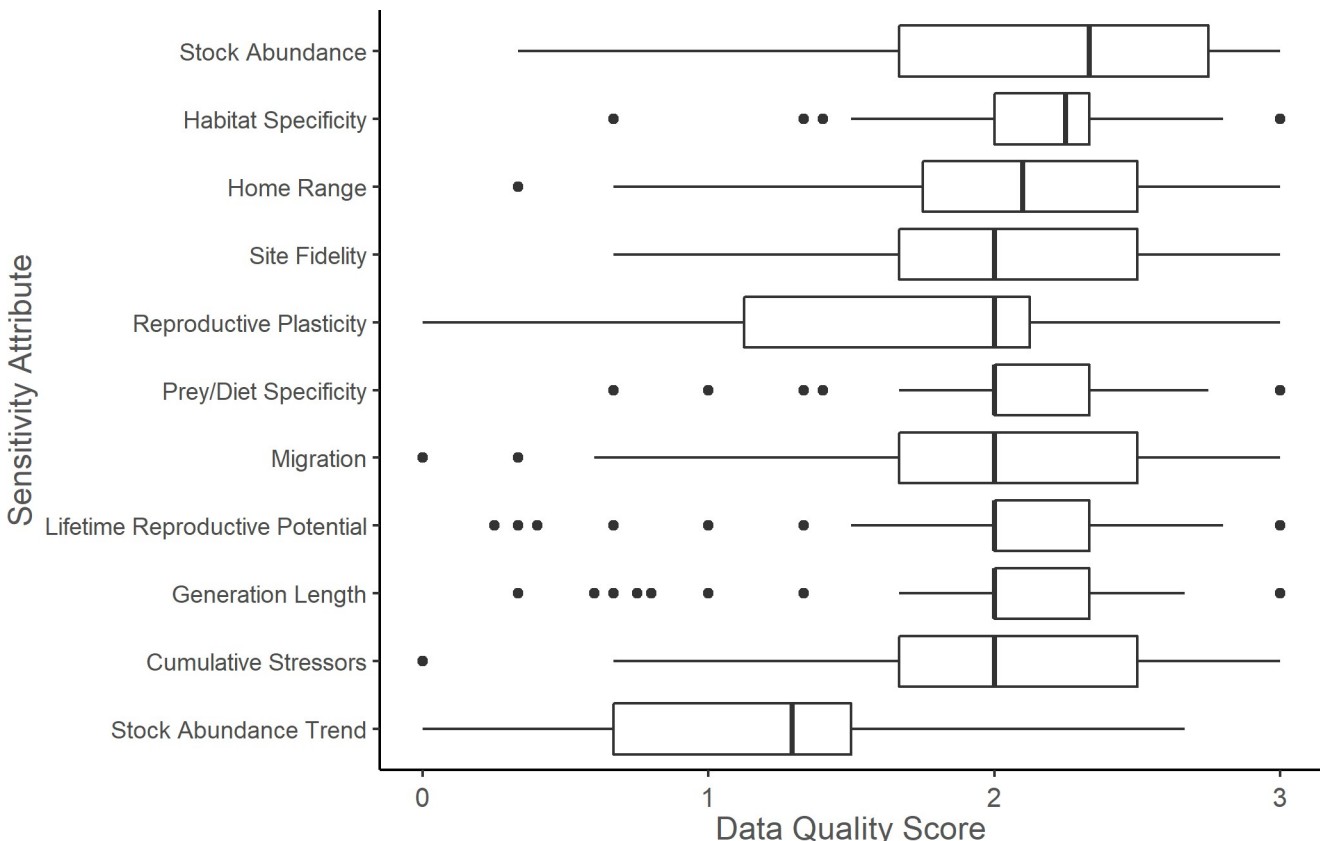

**Fig 8. Mean sensitivity attribute data quality scores.** Mean data quality scores of climate sensitivity attributes for 108 marine mammal stocks in the western North Atlantic, Gulf of Mexico, and Caribbean Sea. The vertical bar represents the median; the box is bounded by the first and third quartiles; whiskers represent 1.5 times the inter-quartile range; points represent all outlying values.

small amount of warming SST. Furthermore, stock distributions limited by historical or current human activity may affect the exposure scoring by mischaracterizing the stock extent, particularly for stocks that are recovering and repopulating areas that had been previously abandoned [141]. The stocks evaluated in this assessment covered a wide range of geographic regions where climate impacts are not spatially homogeneous. Thus, exposure factors scored differently based on the geographic location of a given stock. The impacts of climate change were similar for stocks that overlap spatially and any differences in exposure factor scores between stocks were primarily the result of stocks having different geographic distributions.

## Sensitivity

Assessing sensitivity to climate change allows us to understand the intrinsic stock attributes contributing to vulnerability. These sensitivity scores consider what is currently known about the stock and do not project potential changes in life history or ecology in response to climate change. The most influential attributes varied by taxonomic group and, although stocks within this region were dominated by delphinid stocks, some attributes were important across multiple taxonomic groups. Based on the leave-one-out analysis and scores aggregated across all stocks, the most influential sensitivity attributes included migration, site fidelity, home range, and stock abundance. Migration was the most difficult sensitivity attribute for which to establish scoring criteria. Migratory species are often considered to be vulnerable to climate change

due to a reliance on habitat that may be seasonal or ephemeral in nature, such as sea ice or conditions that drive plankton blooms [73, 142, 143]. In this assessment, habitat specificity and site fidelity were considered separately as independent attributes, but the potential for temporal mismatches between migratory stocks and specific environmental conditions remains an important element of migration [66, 69, 73, 75, 144]. The relationship between migratory cues and changing climate conditions may determine whether a migratory species is more or less sensitive to climate-driven shifts in phenology [145]. Cascading impacts of climate change may affect predators, prey, and habitat throughout the migratory cycle and impacts to an area while the stock is not present (e.g., impacts to breeding grounds while the stock is on the foraging grounds) may have downstream effects on the stock once it arrives. Other CVA frameworks have used migration as a proxy for exposure (e.g., impacts in regions outside of the study area [66, 146]) or adaptive capacity (e.g., dispersal ability [143, 144]), which was not necessary with our approach because those elements were explicitly considered within the method. While we also considered the home range of individuals of a stock, the fact that a stock undergoes a long-distance migration and the diversity of the pathways the stock uses within and between years may suggest a relatively inflexible seasonal routine or, alternatively, confer a degree of adaptive capacity [147, 148].

The influence of site fidelity, home range, and stock abundance was more straightforward. If individuals return to a site that is impacted by climate change, those individuals would also be expected to be impacted [73]. Other CVA frameworks have estimated geographic extent using metrics such as discrete area [69, 149], latitudinal range [66, 150] or longitudinal range [73]. With a strong imbalance in available geographic range information between stocks, our approach used generalized terms to define home range, with an emphasis on habitat connectivity. Species with individuals that have a broad home range may be able to avoid patches of unfavorable conditions within their home range and find other suitable habitat more easily than species with individuals with a narrow home range. This was particularly evident in the differences observed between the bay, sound, and estuary common bottlenose dolphin stocks that scored very high vulnerability and the coastal/offshore common bottlenose dolphin stocks that generally scored low, moderate, and high vulnerability. Stocks with low abundance generally have less diversity (e.g., genetic, behavioral, cultural) and are at greater risk of disturbance and extinction [151–153] while stocks with high abundance generally have greater diversity and may be better suited to expand their range or re-establish in in formerly occupied areas[73, 84].

In some instances, sensitivity attribute scores could be artificially pushed towards less sensitive bins due to human influence on the species. Most notably, this occurred for the North Atlantic right whale with the generation time attribute. North Atlantic right whale abundance has decreased due to human-induced mortality, particularly from vessel strikes and entanglements [154–157]. Because these mortalities often occur prior to the individuals reaching the expected age at final reproduction [158], the generation length of this stock is shortened and, therefore, scores as lower sensitivity compared with what would otherwise be expected without that human-induced mortality. Additionally, shifting prey distributions and phenology have been shown to affect the North Atlantic right whale, which may affect this species' survival [159–161].

Cumulative stressors were a major contributor to the vulnerability of many stocks. Among the sensitivity attributes scored, non-climate stressors have the greatest potential to be addressed and managed, and several species and stocks are currently at risk of extinction or regional extirpation from non-climate stressors [162]. Stressors that threaten recovery, and otherwise cause mortality or sub-lethal impacts, include bycatch and entanglement [163–165], vessel strikes [166], habitat degradation [167–169], increasing anthropogenic sound in the environment [170–172], chemical pollution [173–175], marine debris [176, 177], disease [178, 179], and harmful algal blooms [180–182]. While these other stressors were jointly considered

as part of the "cumulative stressors" attributes, there could be important synergistic effects between stressors or between stressors and climate that influence vulnerability and further work should be done to incorporate these impacts as well.

Although all marine mammal stocks in the United States are protected under the MMPA, those stocks that are afforded additional protection by being designated "endangered" under the ESA or designated as "depleted" or "strategic" under the MMPA also appear to be among the most vulnerable to climate change (e.g., Rice's whale [*Balaenoptera ricei*], North Atlantic right whale). Many of the same attributes that cause a stock to be afforded additional regulatory protections are the same attributes that increase the stocks' vulnerability to climate change.

## Comparison to other assessments

Similar trait-based CVAs were used for a regional assessment of cetaceans in the Madeira Archipelago off the coast of Portugal in the northeast Atlantic [74], and for a global assessment of marine mammals conducted at the species level [76]; however, methodological differences between those two studies and ours make direct comparisons difficult to interpret. A study of cetacean climate vulnerability in Macaronesia adopted the same method used in our study [75].

All species in our study were included in Albouy et al. [76], although our study further divided the assessment units to the stock level for some species, with multiple stocks for most species. Our study and the Sousa et al. [74, 75] studies assessed sperm whales, fin whales (*Balaenoptera physalus*), common bottlenose dolphins, short-finned pilot whales (*Globicephala macrorhynchus*), common dolphins, and Atlantic spotted dolphins (*Stenella frontalis*), but the populations assessed in both studies are found in different regions of the North Atlantic. Results at the species-level were in general agreement; however, there were noticeable differences in vulnerability for some species. These differences in results between studies highlight the importance of defining the assessment unit, as intraspecific variation may drive more severe responses in smaller organizational units [183]. In the United States, marine mammals are managed at the level of the stock, which are most typically finer-scale management units than species [98]. Using stocks as an assessment unit provided a level of information resolution that can be directly applicable to management agencies and restoration plans. Common bottlenose dolphin is presented below as one example of a species with differing results between assessments.

Albouy et al. [76] and Sousa et al. [74] both found common bottlenose dolphins to score among the lower range of vulnerability and sensitivity to climate change, although Sousa et al. [74] found the island-associated common bottlenose dolphin population to have slightly higher sensitivity relative to the offshore common bottlenose dolphin population. Using the same method as our study, Sousa et al. [75] found island-associated common bottlenose dolphins to be very high vulnerability and offshore common bottlenose dolphins to be high vulnerability. In our study, bay, sound, and estuary common bottlenose dolphin stocks along the GOMx and WNA coasts scored high and very high vulnerability, respectively. This was primarily in consideration of their small home ranges, non-migratory behavior, and small stock abundances (e.g., [184–189]). The site fidelity and ranging patterns of some of these stocks have contributed to their vulnerability to other environmental disturbances not related to climate change [190, 191]. Furthermore, some of these stocks have not shown range shifts to date in the face of environmental catastrophes such as hurricanes and oil spills (e.g., [182, 192–195]). Coastal and offshore common bottlenose dolphin stocks in the GOMx and WNA have larger home ranges, more migratory behavior, and larger population abundances (e.g., [196–201]), resulting in scores of moderate or high vulnerability instead of very high vulnerability for these stocks.

## Assessment design and future improvements

To be truly useful for management purposes, trait-based CVAs require assessment of their sensitivity and uncertainty [60]. In our study, sources of variability included inter-scorer interpretation of information and variability of underlying experience and knowledge. To account for this variability among scorers, we ran a bootstrap analysis [70, 75, 84]. We also conducted a leave-one-out sensitivity analysis to assess the effect of each attribute on the overall vulnerability scores [202]. We conducted a similar leave-one-out analysis by sequentially calculating the vulnerability scores with a single expert's scores removed to estimate the effect each expert had on scores. We did not encourage experts to work toward consensus during the scoring discussion portion of the process, and variability between experts was expected. One approach to reduce the effect of individual scorers would be to increase the number of scorers per stock. The number of scorers per stock is limited by expert availability, and there are tradeoffs between increasing the number of scorers, increasing individual scorer load, and maintaining a manageable number of scorers for group dynamics and logistical purposes. Another strategy to account for variability among scorers would be to repeat the assessment with a separate set of scorers; however, the effect of selecting a separate set of scorers in trait-based CVAs has not yet been tested. Sources of uncertainty included availability and quality of underlying data, scoring criteria thresholds, and logic model thresholds. We accounted for uncertainty in the availability and quality of underlying data with a data quality score. Although the data quality score did not have an influence on the final vulnerability score, it provided additional context about which scores have the greatest confidence. There are species and stocks included in this assessment that are understudied (e.g., beaked whale species, ziphiid group), particularly with respect to sensitivity attributes. Although some of the stocks considered here are transboundary (e.g., North Atlantic right whale, harbor porpoise, gray seal, most pelagic species in the Gulf of Mexico), much of the data available for these stocks are from United States and Canadian waters where survey effort may be spatially and/or temporally uneven, and data availability beyond territorial waters is even sparser [101, 203–205]. Even with features that allow trait-based approaches to function in cases where there is a lack of data, such as data quality criteria and the ability of scorers to spread tallies [59], data gaps exist that need to be filled to minimize uncertainty, and the assessment could certainly be improved with more underlying data. For example, the ziphiid group had lower median data quality scores for all sensitivity attributes than each of the other taxonomic groups. Until such time that those data gaps can be filled, these CVA results should be interpreted with data quality scores and uncertainty metrics in mind. Future analyses should be conducted when additional data become available.

Applications of expert elicitation can often leave questions about influence and bias [206, 207]. We used a Delphi approach to minimize biases such as groupthink (a form of consensus seeking due to social pressure), deference to authority (social pressure to agree with more senior or more experienced experts), and halo effect (considering scoring criteria beyond those which were provided) that may be more prevalent with other types of expert elicitation [85, 87, 207–209]. Using an approach in which experts first scored individually before later discussing as a group reduced groupthink and deference to authority by allowing each expert to establish their own score independent of others. Any scoring adjustments occurred independently following the group discussions and we reiterated that consensus was not a goal of the discussions. The group discussions contributed to minimizing halo effect by allowing scorers to identify reasoning that may fall outside the guidelines of the criteria.

One criticism of trait-based CVAs is that scoring thresholds can seem arbitrary [57, 210]. The MMCVA scoring thresholds were designed to produce meaningful separation among marine mammal stocks, and descriptive scoring criteria were used to improve transparency of

the underlying scoring process. The inclusion or omission of specific life history traits affects overall performance of a vulnerability assessment [70, 202]. For example, social structure is an important aspect of certain marine mammal populations and the degree to which climate change may affect species' social structure or how social structure may impact sensitivity or adaptive capacity is under continued study [211, 212]. We were unable to determine an appropriate scoring scheme for social structure in this assessment but encourage its inclusion in future iterations. We also did not discuss potential human responses to climate change (e.g., shipping patterns, fishing effort, coastal development trends) or potential shifts in prey distribution [213, 214]. Additionally, measures to mitigate rising sea levels (e.g., shoreline hardening, river diversions, etc.) may adversely affect marine mammals and their prey, and these human response factors are worth considering in future analyses.

We recognize that changes in the logic model criteria could result in changes in vulnerability scores, but we chose to use the logic model established in the FCVA [70, 84] to maintain consistency with other NOAA Fisheries CVAs. Due to the logic model, factors and attributes with the highest scores also had the greatest impact when removed during the sensitivity analysis.

Lower sensitivity component scores resulted from removing attributes from the model, suggesting that adaptive capacity is not fully integrated into the scores, because removing an adaptive capacity attribute would have the expected result of increasing the sensitivity component score. We combined sensitivity and adaptive capacity into a single component to avoid the potential for considering a single life history attribute in two places. Future iterations of marine mammal CVAs might include the explicit consideration of a separate adaptive capacity component [215], as was done with Australian lizard species [216]. Those authors found adaptive capacity alone had the potential to shift vulnerability scores lower. Also, incorporating the results of prey and habitat vulnerability assessments to better show cascading effects should be considered in future studies. Evidence of adaptive capacity is important to document, and when combined with the results of prey and habitat vulnerability assessments, future iterations may have the potential to signal cascading ecological effects.

Automating exposure scoring in a geographical information system (GIS) could improve scoring accuracy and efficiency but was confounded by imprecise stock boundaries. Future iterations are encouraged to explore using density models to score exposure.

We used climate projections from the NOAA Climate Change Web Portal that had a spatial resolution of 1 degree by 1 degree, which is larger than the range of some stocks we scored, namely the common bottlenose dolphin bay, sound, and estuary stocks. Some of those common bottlenose dolphin bay, sound, and estuary stocks (e.g., Sabine Lake, Sarasota Bay, Caloosahatchee River) occupy an area less than 0.5 degree by 0.5 degree [217, 218]. Downscaled or regional climate models have greater capability to resolve fine-scale features, which are poorly resolved by the global climate models. Although the coarse-resolution models used in our assessment poorly resolve changes in areas such as the Gulf of Maine [21] and inshore areas, the ability to compare exposure across stocks and regions outweighed the improvements in accuracy for specific locations [202, 219]. Downscaled models across the region would improve the accuracy of the exposure scores and should be considered in future studies as downscaled models become available for more areas.

## Recommendations and conclusions

Marine mammal stocks are experiencing impacts from climate change and are expected to respond with shifting distribution, changes in abundance, and/or changing phenology. These changing climate conditions, and the potential responses of marine mammal stocks to these

changing conditions, pose challenges to the management of these stocks. Our assessment was designed to provide climate vulnerability information as an initial step to inform marine mammal management under changing climate conditions. This assessment identified the stocks most vulnerable to climate change, which can help to identify those that should be prioritized for monitoring and advanced modeling to predict and detect changes in distribution, abundance, and phenology [46]. The assessment should be repeated once new CVA input information becomes available, such as climate projections with higher resolution and greater confidence produced by next-generation climate models and/or new stock-specific biological information, particularly for data-limited stocks. The results from this assessment can help advance research into marine mammal responses to climate change and inform the management and recovery of these stocks under changing climate conditions. This vulnerability assessment provides a tool that can complement other marine mammal assessment techniques and support the broader implementation of protected species and ecosystem management and conservation as the climate changes.

## Supporting information

**S1 Dataset. Raw attribute and factor scores.** Expert scores for each exposure factor and sensitivity attribute aggregated by stock.
(CSV)

**S1 File. Stock profiles and narratives.** Detailed scoring results profile and descriptions of the climate impacts on each stock, including key drivers of vulnerability, exposure, and sensitivity; data quality and gaps; and background information on the stock.
(PDF)

**S2 File. Sensitivity attribute and exposure factor mean scores summarized by taxonomic group.** Box and whisker plots of sensitivity attribute mean scores and exposure factor mean scores for each of the five taxonomic groups.
(DOCX)

**S3 File. Expert effect on scores.** Leave-one-out analysis for expert scores.
(DOCX)

**S1 Table. Stocks scored.** List of all marine mammal stocks and stock groupings in the assessment.
(XLSX)

**S2 Table. Stock exposure, sensitivity, and vulnerability results summary table.**
(XLSX)

## Acknowledgments

We thank the members of the Protected Species Climate Vulnerability Assessment (PSCVA) steering committee for their guidance on this project. We thank the PSCVA expert workshop participants for their input on assessment framework and initial attribute lists. We thank NOAA Fisheries Office of Science and Technology and Office of Protected Resources leadership for their critical support throughout the project. We thank the NOAA Fisheries Protected Resources Board for their feedback and support. We thank Megan Stachura for providing R scripts used in other NOAA vulnerability assessments. We thank Mark Nelson for his guidance and advice throughout the project. We thank Mike Johnson and Teri Rowles for their reviews of the manuscript. Acknowledgment of the above individuals and groups does not

imply their endorsement of this work; the authors have sole responsibility for the content of this contribution.

The views expressed herein are the authors' and do not necessarily reflect the views of NOAA or any of its sub-agencies.

## Author Contributions

**Conceptualization:** Matthew D. Lettrich, Michael J. Asaro, Diane L. Borggaard, Dorothy M. Dick, Roger B. Griffis, Jenny A. Litz, Christopher D. Orphanides, Debra L. Palka, Melissa S. Soldevilla.

**Data curation:** Matthew D. Lettrich.

**Formal analysis:** Matthew D. Lettrich.

**Investigation:** Matthew D. Lettrich, Jenny A. Litz, Christopher D. Orphanides, Debra L. Palka, Melissa S. Soldevilla, Brian Balmer, Samuel Chavez, Danielle Cholewiak, Diane Claridge, Ruth Y. Ewing, Kristi L. Fazioli, Dagmar Fertl, Erin M. Fougeres, Damon Gannon, Lance Garrison, James Gilbert, Annie Gorgone, Aleta Hohn, Stacey Horstman, Beth Josephson, Robert D. Kenney, Jeremy J. Kiszka, Katherine Maze-Foley, Wayne McFee, Keith D. Mullin, Kimberly Murray, Daniel E. Pendleton, Jooke Robbins, Jason J. Roberts, Grisel Rodriguez- Ferrer, Errol I. Ronje, Patricia E. Rosel, Todd Speakman, Joy E. Stanistreet, Tara Stevens, Megan Stolen, Reny Tyson Moore, Nicole L. Vollmer, Randall Wells, Heidi R. Whitehead, Amy Whitt.

**Methodology:** Matthew D. Lettrich, Michael J. Asaro, Diane L. Borggaard, Dorothy M. Dick, Roger B. Griffis, Jenny A. Litz, Christopher D. Orphanides, Debra L. Palka, Melissa S. Soldevilla, Daniel E. Pendleton.

**Project administration:** Matthew D. Lettrich.

**Validation:** Michael J. Asaro, Diane L. Borggaard, Dorothy M. Dick, Roger B. Griffis, Jenny A. Litz, Christopher D. Orphanides, Debra L. Palka, Melissa S. Soldevilla, Brian Balmer, Samuel Chavez, Danielle Cholewiak, Diane Claridge, Ruth Y. Ewing, Kristi L. Fazioli, Dagmar Fertl, Erin M. Fougeres, Damon Gannon, Lance Garrison, James Gilbert, Annie Gorgone, Aleta Hohn, Stacey Horstman, Beth Josephson, Robert D. Kenney, Jeremy J. Kiszka, Katherine Maze-Foley, Wayne McFee, Keith D. Mullin, Kimberly Murray, Daniel E. Pendleton, Jooke Robbins, Jason J. Roberts, Grisel Rodriguez- Ferrer, Errol I. Ronje, Patricia E. Rosel, Todd Speakman, Joy E. Stanistreet, Tara Stevens, Megan Stolen, Reny Tyson Moore, Nicole L. Vollmer Randall Wells, Heidi R. Whitehead, Amy Whitt.

**Visualization:** Matthew D. Lettrich.

**Writing – original draft:** Matthew D. Lettrich.

**Writing – review & editing:** Matthew D. Lettrich, Michael J. Asaro, Diane L. Borggaard, Dorothy M. Dick, Roger B. Griffis, Jenny A. Litz, Christopher D. Orphanides, Debra L. Palka, Melissa S. Soldevilla, Brian Balmer, Samuel Chavez, Danielle Cholewiak, Diane Claridge, Ruth Y. Ewing, Kristi L. Fazioli, Dagmar Fertl, Erin M. Fougeres, Damon Gannon, Lance Garrison, James Gilbert, Annie Gorgone, Aleta Hohn, Stacey Horstman, Beth Josephson, Robert D. Kenney, Jeremy J. Kiszka, Katherine Maze-Foley, Wayne McFee, Keith D. Mullin, Kimberly Murray, Daniel E. Pendleton, Jooke Robbins, Jason J. Roberts, Grisel Rodriguez- Ferrer,

Errol I. Ronje, Patricia E. Rosel, Todd Speakman, Joy E. Stanistreet, Tara Stevens,
Megan Stolen, Reny Tyson Moore, Nicole L. Vollmer, Randall Wells, Heidi R. Whitehead,
Amy Whitt.

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
