## [Decision Letter · Decision Letter 0]

29 Mar 2023

PONE-D-23-05946Vulnerability to climate change of United States marine mammal stocks in the western North Atlantic, Gulf of Mexico, and CaribbeanPLOS ONE

Dear Dr. Lettrich,

Thank you for submitting your manuscript to PLOS ONE. After careful consideration, we feel that it has merit but does not fully meet PLOS ONE’s publication criteria as it currently stands. Therefore, we invite you to submit a revised version of the manuscript that addresses the points raised during the review process.

We received comments from two reviewers. They are both positive about the work. They expressed some concerns. Please address these concerns in the revisions. My opinion toward the work is slightly more negative than the reviewers (please see my separate comments), but I respect the opinions of the two reviewers. So I plan to ask the same reviewers to review a revised manuscript, first (up to two reviewers) in the next round of reviews. Then, I will decide solely based on the recommendations.

We look forward to receiving your revised manuscript.

Kind regards,

Masami Fujiwara, PhD

Academic Editor

PLOS ONE

Journal Requirements:

https://spo.nmfs.noaa.gov/sites/default/files/TMSPO196_508.pdf

In your revision ensure you cite all your sources (including your own works), and quote or rephrase any duplicated text outside the methods section. Further consideration is dependent on these concerns being addressed

Additional Editor Comments (if provided):

The study can be viewed as a special type of meta-analysis. In a typical meta-analysis, each study is treated as one sample whereas the current study treats one researcher as a sample. Of course, treating one study as one sample has a problem; we all know not all studies are created equal. However, treating one reviewer as one sample also has a problem (they are probably not independent; see comments below). I also think the selection of researchers can lead to bias in the results. In the past, I have experience working with marine mammal scientists. Some of them have strong opinions about what they “like” and what they do “not like” about study results. A large amount of subjectivity is unavoidable in scoring (i.e. there is a filter going from data to scores). It would have been more convincing if the scoring was done by experts in population dynamics without much knowledge of marine mammals. I like the part the study also scores the data quality, but there is still a large uncertainty in how those data are interpreted by the researchers. This needs to be acknowledged. The tone of the language in the abstract and discussion suggesting a high level of confidence is not supported by the study results. In general, a large uncertainty in the results of the study is not quantified sufficiently.

The researchers are treated as independent samples. But I doubt they are. If they work closely with each other (or their institution’s missions are the same), they are dependent. One of the interesting analyses may be to separate NOAA researchers and others. I noticed there are many NOAA researchers involved in research. I am wondering if the results are different between the two groups. If not, perhaps, we do not worry about the institutional dependency. One may be able to group the research based on the taxa of their expertise. I also suggest bootstrapping researchers for all species at the same time to investigate the bias of certain researchers. Some tend to score low (or high) for all species. These and other potential dependencies of researchers will influence the overall results (potentially leading to over-confidence as well as bias in the results). Further investigation of dependency might make the work slightly more convincing.

Reviewers' comments:

Reviewer's Responses to Questions

**Comments to the Author**

1. Is the manuscript technically sound, and do the data support the conclusions?

Reviewer #1: Yes

Reviewer #2: Yes

2. Has the statistical analysis been performed appropriately and rigorously? 

Reviewer #1: Yes

Reviewer #2: Yes

3. Have the authors made all data underlying the findings in their manuscript fully available?

Reviewer #1: Yes

Reviewer #2: Yes

4. Is the manuscript presented in an intelligible fashion and written in standard English?

Reviewer #1: Yes

Reviewer #2: No

5. Review Comments to the Author

Reviewer #1: Summary:

This manuscript uses an expert solicitation to assess the vulnerability of 108 US East Coast / Gulf of Mexico / Caribbean marine mammal stocks to climate change. Authors assess vulnerability using a robust model that accounts for a suite of physical variables, stock-specific life history traits and stock-specific spatial distribution. Data quality for each stock and variation in responses between experts are considered and presented as well. This work is helpful for identifying which stocks will need targeted conservation and management effort, and also provides a baseline to compare future evaluations of climate vulnerability in these taxa.

General comments:

Some description of the data presented in tables and figures is appropriate, but this could be pared down in the text.

When aggregating results across all 108 stocks, it would be good to remind the readers that these trends will be more representative of delphinids, since they are the majority of the stocks analyzed.

This manuscript needs more clarity on how physical variables were treated spatially, and how the spatial representation of these variables were paired up with the geographic distribution of each stock.

Specific comments:

Line 106: “for others” – do you mean “in other regions”?

Line 107: Suggest changing “For example” to “In another example”

Line 130: Not sure what is meant by “habitats” in this sentence.

Lines 173-200 and Table 1: How were these scores for physical variables (SST, dissolved oxygen, etc.) treated differently for each marine mammal stock based on its geographic distribution? For example, when you are evaluating a marine mammal stock in the GOMx, you wouldn’t include SST from the Gulf of Maine, right? You would need to provide SST metrics (change in mean and change in variability) that were calculated over that stock’s spatial distribution.

Line 304: Here it is still confusing how the physical variables were aggregated spatially to be used in comparison with the species range maps.

Line 344: I don’t believe “certainty” is an appropriate word here (and in the associated results text). The certainty of your vulnerability scores is intrinsically related to the quality of the data and the expertise and objectivity of the expert participants, neither of which seem to be incorporated into these scores. “Score variance” may be more appropriate here.

Fig. 2 seems redundant with Fig. 1

Fig. 3: Please vary the scales along the x axis so that we can see the scores for the non-delphinid taxonomic groups

Lines 426-443: This section literally describes the exact numbers shown in Fig. 3. It’s ok to draw readers attention to the most important findings but, as written, much of this text seems redundant and takes up journal space.

Line 469: Can you add a sentence here, and perhaps in the figure caption, clarifying that the other attributes did not impact vulnerability scores when they were removed from the analysis? The figure almost looks like there is missing data, but I think these attributes just don’t impact vulnerability.

Line 485: Are these unformatted citations?

Lines 479-486: This section is biased by the taxonomic skew among the stocks represented in this study. For example, since most of the stocks are dolphins, they dominate the results in this section. I think this analysis would make more sense if it was partitioned by the 5 taxonomic groups

Line 578: “most influential” – is this biased to refer to the delphinid taxa?

Line 630: I would include “entanglement” explicitly, but you could group this with bycatch by saying “bycatch and entanglement” if you prefer.

Line 701-702: Do you mean “data quality scores” here? This is unclear

Reviewer #2: This manuscript describes a climate-variability assessment of numerous marine mammal stocks in three different regions using expert elicitation. The manuscript was well written and clearly the product of a lot of effort by the authors. As I am not overly familiar with the methods used in the paper, I particularly appreciated the use of examples throughout to help clarify the methods and interpretation of the results. My few comments below are minor and easily addressed by the authors. My 'biggest' criticism is that the Figures 1 -3 convey the main findings of the paper but they are not very visually appealing. I think it does the manuscript a disservice, particularly given that this is a fairly broad paper that could be of considerable interest, and that some relatively changes would make these figures more impactful. Overall though a really interesting and comprehensive manuscript that I look forward to seeing published.

Specific comments:

Table 1: Most of these are the same, just being repeated for different variables. To condense, I would suggest having one row each for the change in mean and change in variability for all the variables except circulation and sea level rise, and then separate rows for circulation and sea level rise since the criteria differed for these two variables

Figure 3: One of your headings is cut off. I would also suggest using different scales across the column, as bars for non-delphinid groups are difficult to see because of differences in the number of stocks.

Line 245: I would suggest removing the NARW abbreviation here. I realize it comes up again in the discussion, but since there is such a large gap between the first mention and the second mention, and that the overall number of mentions are relatively small, I don't think it adds more confusion to use an abbreviation.

Figures: It would be nice of there at least one figure that summarized the results in a very visually appearing nature. Perhaps if possible in some sort of map format, although I recognize this may be too challenging given the range differences in some species

6. PLOS authors have the option to publish the peer review history of their article (what does this mean?). If published, this will include your full peer review and any attached files.

Reviewer #1: **Yes: **Erin Meyer-Gutbrod

Reviewer #2: No

---

## [Author Response · Author response to Decision Letter 0]

7 Jul 2023

Reviewer #1

Summary:

This manuscript uses an expert solicitation to assess the vulnerability of 108 US East Coast / Gulf of Mexico / Caribbean marine mammal stocks to climate change. Authors assess vulnerability using a robust model that accounts for a suite of physical variables, stock-specific life history traits and stock-specific spatial distribution. Data quality for each stock and variation in responses between experts are considered and presented as well. This work is helpful for identifying which stocks will need targeted conservation and management effort, and also provides a baseline to compare future evaluations of climate vulnerability in these taxa.

General comments:

Some description of the data presented in tables and figures is appropriate, but this could be pared down in the text.

**We reduced text in the results sections (Lines 432–440).

When aggregating results across all 108 stocks, it would be good to remind the readers that these trends will be more representative of delphinids, since they are the majority of the stocks analyzed.

**We have added text to note the stock proportions and also broken out results by taxonomic group where appropriate.

This manuscript needs more clarity on how physical variables were treated spatially, and how the spatial representation of these variables were paired up with the geographic distribution of each stock.

**We addressed the specific in-line comments below related to this.

Specific comments:

Line 106: “for others” – do you mean “in other regions”?

**Updated to read “for other parameters” instead of “for others.” (Lines 108–109)

Line 107: Suggest changing “For example” to “In another example”

**Suggestion incorporated into the text. (Lines 109–112)

Line 130: Not sure what is meant by “habitats” in this sentence.

**Updated to read “marine and coastal habitat” instead of “habitats.” (Lines 131–134)

Lines 173-200 and Table 1: How were these scores for physical variables (SST, dissolved oxygen, etc.) treated differently for each marine mammal stock based on its geographic distribution? For example, when you are evaluating a marine mammal stock in the GOMx, you wouldn’t include SST from the Gulf of Maine, right? You would need to provide SST metrics (change in mean and change in variability) that were calculated over that stock’s spatial distribution.

**In lines 175–176, we indicate that exposure for each stock is scored within the bounds of its current geographic distribution. To further clarify this, we modified the text in the “Scoring Approach” section (Lines 308–312): “Experts compared the geographic distribution of each stock (based on range maps, sighting data, and density plots from the published literature) to the projected exposure level for each individual factor and scored each exposure factor by placing five tallies across the four bins according to the magnitude of exposure projected across the entirety of the stock’s current distribution.”

Line 304: Here it is still confusing how the physical variables were aggregated spatially to be used in comparison with the species range maps.

**Please see the response immediately above.

Line 344: I don’t believe “certainty” is an appropriate word here (and in the associated results text). The certainty of your vulnerability scores is intrinsically related to the quality of the data and the expertise and objectivity of the expert participants, neither of which seem to be incorporated into these scores. “Score variance” may be more appropriate here.

**The term “certainty” has been used in this capacity in other CVAs that used similar methodology (Hare et al. 2016, Spencer et al. 2019, Crozier et al. 2019, Farr et al. 2021, Giddens et al. 2022, McClure et al. 2023). We have added text (Lines 345–347) explaining “In this context, certainty is a measure of the consistency of the original score with the bootstrap score. The bootstrap score provides an estimate of the vulnerability score sensitivity given the variance in the scoring distribution.” 

Fig. 2 seems redundant with Fig. 1

**Fig. 2 has been removed and all other figure numbers updated accordingly

Fig. 3: Please vary the scales along the x axis so that we can see the scores for the non-delphinid taxonomic groups

**Scales were updated to allow better resolution and the caption was updated to draw attention to the varying scales. (Lines 441–442)

Lines 426-443: This section literally describes the exact numbers shown in Fig. 3. It’s ok to draw readers attention to the most important findings but, as written, much of this text seems redundant and takes up journal space.

**We reduced the text in this section to highlight only specific findings for each taxonomic group. (Lines 430–438)

Line 469: Can you add a sentence here, and perhaps in the figure caption, clarifying that the other attributes did not impact vulnerability scores when they were removed from the analysis? The figure almost looks like there is missing data, but I think these attributes just don’t impact vulnerability.

**Added text: “No other factors changed vulnerability scores when omitted during the leave-one-out analysis.” (Lines 466–467)

Added text to figure caption: “Only ocean pH (change in mean), dissolved oxygen (change in mean), air temperature (change in mean), and sea surface temperature (change in mean) changed vulnerability scores when omitted.” (Lines 475–477)

Line 485: Are these unformatted citations?

**Yes, thank you for identifying this error. Citations have been updated in the text. (Line 489)

Lines 479-486: This section is biased by the taxonomic skew among the stocks represented in this study. For example, since most of the stocks are dolphins, they dominate the results in this section. I think this analysis would make more sense if it was partitioned by the 5 taxonomic groups

**We added taxonomic group-specific plots to a new supplementary material section (S5. Taxonomic group attribute and factor scores).

We added the following text: “Migration was the attribute with the highest median score for 3 of the taxonomic groups (delphinid, ziphiid, and other odontocete) while the prey and diet specificity attribute had the highest median score for the mysticete group and habitat specificity had the highest median score for the pinniped group (S5).” (Lines 481–484)

Additionally, we added parenthetical text to describe results by taxonomic group.

Line 578: “most influential” – is this biased to refer to the delphinid taxa?

**Yes. We added text to indicate that the most influential attributes varied by taxonomic group and although stocks within this region were dominated by delphinid stocks, some attributes were important across multiple taxonomic groups (Lines 591–595).

Line 630: I would include “entanglement” explicitly, but you could group this with bycatch by saying “bycatch and entanglement” if you prefer.

**Changed text to “bycatch and entanglement” (Line 644)

Line 701-702: Do you mean “data quality scores” here? This is unclear

**Added text to indicate we are referring to “lower median data quality scores.” (Line 725)

Reviewer #2:

This manuscript describes a climate-variability assessment of numerous marine mammal stocks in three different regions using expert elicitation. The manuscript was well written and clearly the product of a lot of effort by the authors. As I am not overly familiar with the methods used in the paper, I particularly appreciated the use of examples throughout to help clarify the methods and interpretation of the results. My few comments below are minor and easily addressed by the authors. My 'biggest' criticism is that the Figures 1 -3 convey the main findings of the paper but they are not very visually appealing. I think it does the manuscript a disservice, particularly given that this is a fairly broad paper that could be of considerable interest, and that some relatively changes would make these figures more impactful. Overall though a really interesting and comprehensive manuscript that I look forward to seeing published.

Specific comments:

Table 1: Most of these are the same, just being repeated for different variables. To condense, I would suggest having one row each for the change in mean and change in variability for all the variables except circulation and sea level rise, and then separate rows for circulation and sea level rise since the criteria differed for these two variables

**Please see the updated Table 1 with the reviewer’s suggestion incorporated

Figure 3: One of your headings is cut off. I would also suggest using different scales across the column, as bars for non-delphinid groups are difficult to see because of differences in the number of stocks.

**The heading has been updated and the scales have been adjusted.

Line 245: I would suggest removing the NARW abbreviation here. I realize it comes up again in the discussion, but since there is such a large gap between the first mention and the second mention, and that the overall number of mentions are relatively small, I don't think it adds more confusion to use an abbreviation.

**We changed all occurrences of “NARW” to “North Atlantic right whale.”

Figures: It would be nice of there at least one figure that summarized the results in a very visually appearing nature. Perhaps if possible in some sort of map format, although I recognize this may be too challenging given the range differences in some species

**We explored visualizing the results in a map-based format however the number of stocks and the diverse geographic boundaries of the stocks made the map difficult to read.

Editor:

We noticed you have some minor occurrence of overlapping text with the following previous publication(s), which needs to be addressed:

https://spo.nmfs.noaa.gov/sites/default/files/TMSPO196_508.pdf

**Overlapping text was identified in the Discussion section and rephrased.

In your revision ensure you cite all your sources (including your own works), and quote or rephrase any duplicated text outside the methods section. Further consideration is dependent on these concerns being addressed

The study can be viewed as a special type of meta-analysis. In a typical meta-analysis, each study is treated as one sample whereas the current study treats one researcher as a sample. Of course, treating one study as one sample has a problem; we all know not all studies are created equal. However, treating one reviewer as one sample also has a problem (they are probably not independent; see comments below). I also think the selection of researchers can lead to bias in the results. In the past, I have experience working with marine mammal scientists. Some of them have strong opinions about what they “like” and what they do “not like” about study results. A large amount of subjectivity is unavoidable in scoring (i.e. there is a filter going from data to scores). It would have been more convincing if the scoring was done by experts in population dynamics without much knowledge of marine mammals. I like the part the study also scores the data quality, but there is still a large uncertainty in how those data are interpreted by the researchers. This needs to be acknowledged. The tone of the language in the abstract and discussion suggesting a high level of confidence is not supported by the study results. In general, a large uncertainty in the results of the study is not quantified sufficiently.

The researchers are treated as independent samples. But I doubt they are. If they work closely with each other (or their institution’s missions are the same), they are dependent. One of the interesting analyses may be to separate NOAA researchers and others. I noticed there are many NOAA researchers involved in research. I am wondering if the results are different between the two groups. If not, perhaps, we do not worry about the institutional dependency. One may be able to group the research based on the taxa of their expertise. I also suggest bootstrapping researchers for all species at the same time to investigate the bias of certain researchers. Some tend to score low (or high) for all species. These and other potential dependencies of researchers will influence the overall results (potentially leading to over-confidence as well as bias in the results). Further investigation of dependency might make the work slightly more convincing.

**Applications of expert elicitation can often leave questions about influence and bias.

We used a Delphi approach to minimize biases such as groupthink, deference to authority, and halo effect.

Additionally, we used pre-defined criteria for scoring life history traits. Some other CVAs have asked experts to provide their opinion on whether a particular life history trait makes the population more or less vulnerable. To reduce some of the subjectivity associated with those approaches, we had experts score based on how well a population meets the criteria in four separately defined bins for each life history trait. This makes the process more repeatable and provides an opportunity to groundtruth expert scores against published values. The use of these defined bins for each life history trait made the domain-specific knowledge (i.e., marine mammal experts) an important criteria for selecting participants, as they already possessed years of experience with the stocks and an understanding of the literature that would have required extensive training to bring anyone lacking familiarity with marine mammals up to the necessary baseline.

To alleviate concerns about expert effects and institutional influence (i.e. NOAA affiliation), we have included a leave-one-out analysis of the scorers as supplementary material (S6. Expert effect on vulnerability scores) and added text to the Results and Discussion sections.

We added the following text to the Abstract (Lines 93–95): “We quantified sources of uncertainty by bootstrapping vulnerability scores, conducting leave-one-out analyses of individual attributes and individual scorers, and through scoring data quality for each attribute.”

We modified the subheader “Determining attribute importance: Leave-one-out analysis” in the Methods section (Lines 379–384):

“Determining attribute importance and expert effect: Leave-one-out analysis.

We calculated sensitivity and vulnerability scores for each stock by sequentially omitting each sensitivity attribute. We reported the influence of each sensitivity attribute as the change in sensitivity score and vulnerability score by omitting that sensitivity attribute. We conducted a similar analysis for the effect each expert had on vulnerability scores by sequentially omitting the scores of each expert and recalculating each stocks’ vulnerability score.”

We added the following subheader to the Results section (Lines 500–505): “Expert effect

The combination of expert scoring assignments set up 383 scenarios for expert leave-one-out analysis. The effect of removing an individual expert’s scores resulted in no change in vulnerability score in 77% (n=295) of scenarios, a change in vulnerability score of one category (i.e., moving to an adjacent category) in 21% (n=80) of cases, and a change in score of two categories (e.g., moving from low to high) in 2% (n=8) of scenarios (see S6).”

We added the following text to the Discussion section (Lines 701–708): “We conducted a similar leave-one-out analysis by sequentially calculating the vulnerability scores with a single expert’s scores removed to estimate the effect each expert had on scores. We did not encourage experts to work toward consensus during the scoring discussion portion of the process, and variability between experts was expected. One approach to reduce the effect of individual scorers would be to increase the number of scorers per stock. The number of scorers per stock is limited by expert availability, and there are tradeoffs between increasing the number of scorers, increasing individual scorer load, and maintaining a manageable number of scorers for group dynamics and logistical purposes.”

We also highlight the line: “In our study, sources of variability included inter-scorer interpretation of information and variability of underlying experience and knowledge” that appears earlier in the Discussion (Lines 697–698)

---

## [Editor Report · Decision Letter 1]

12 Jul 2023

PONE-D-23-05946R1Vulnerability to climate change of United States marine mammal stocks in the western North Atlantic, Gulf of Mexico, and CaribbeanPLOS ONE

Dear Dr. Lettrich,

Thank you for submitting your manuscript to PLOS ONE. After careful consideration, we feel that it has merit but does not fully meet PLOS ONE’s publication criteria as it currently stands. Therefore, we invite you to submit a revised version of the manuscript that addresses the points raised during the review process.

Overall, I think the revisions were made appropriately in response to the reviewers’ and my comments. However, you responded to my comment as “Applications of expert elicitation can often leave questions about influence and bias. We used a Delphi approach to minimize biases such as groupthink, deference to authority, and halo effect.” I suggest you include the statement in the text along with brief intuitive descriptions of how the method works to reduce the biases.

We look forward to receiving your revised manuscript.

Kind regards,

Masami Fujiwara, PhD

Academic Editor

PLOS ONE
---

## [Author Response · Author response to Decision Letter 1]

8 Aug 2023

Dear Editors:

Thank you for the constructive comments and for the opportunity to revise and resubmit our manuscript. We appreciate the time and effort the reviewers dedicated to their thoughtful comments. We believe that in addressing the concerns of the reviewers and editor that the manuscript is much improved. Specifically, we reduced text in the results section that was redundant with a table, added language describing the geographic extent of exposure scoring, eliminated overlapping text with our NOAA Technical Memorandum outside of the methodology section, added a supplementary information file and associated text breaking out the aggregated scores for each taxonomic group, and added a supplementary information file and associated text addressing the effect of scorers and institution on vulnerability scores.

We added references [206-209] to the reference list, changed reference [76] from the original paper to the author-corrected citation, and made formatting corrections to other references.

Following this letter are the compiled reviewer and editor comments, along with our responses in italics. Line numbers in our responses refer to the clean copy of the manuscript.

Sincerely,

Matthew Lettrich

Reviewer #1

Summary:

This manuscript uses an expert solicitation to assess the vulnerability of 108 US East Coast / Gulf of Mexico / Caribbean marine mammal stocks to climate change. Authors assess vulnerability using a robust model that accounts for a suite of physical variables, stock-specific life history traits and stock-specific spatial distribution. Data quality for each stock and variation in responses between experts are considered and presented as well. This work is helpful for identifying which stocks will need targeted conservation and management effort, and also provides a baseline to compare future evaluations of climate vulnerability in these taxa.

General comments:

Some description of the data presented in tables and figures is appropriate, but this could be pared down in the text.

We reduced text in the results sections (Lines 430–438).

When aggregating results across all 108 stocks, it would be good to remind the readers that these trends will be more representative of delphinids, since they are the majority of the stocks analyzed.

We have added text to note the stock proportions and also broken out results by taxonomic group where appropriate.

This manuscript needs more clarity on how physical variables were treated spatially, and how the spatial representation of these variables were paired up with the geographic distribution of each stock.

We addressed the specific in-line comments below related to this.

Specific comments:

Line 106: “for others” – do you mean “in other regions”?

Updated to read “for other parameters” instead of “for others.” (Lines 108–109)

Line 107: Suggest changing “For example” to “In another example”

Suggestion incorporated into the text. (Lines 109–112)

Line 130: Not sure what is meant by “habitats” in this sentence.

Updated to read “marine and coastal habitat” instead of “habitats.” (Lines 131–134)

Lines 173-200 and Table 1: How were these scores for physical variables (SST, dissolved oxygen, etc.) treated differently for each marine mammal stock based on its geographic distribution? For example, when you are evaluating a marine mammal stock in the GOMx, you wouldn’t include SST from the Gulf of Maine, right? You would need to provide SST metrics (change in mean and change in variability) that were calculated over that stock’s spatial distribution.

In lines 175–176, we indicate that exposure for each stock is scored within the bounds of its current geographic distribution. To further clarify this, we modified the text in the “Scoring Approach” section (Lines 307–311): “Experts compared the geographic distribution of each stock (based on range maps, sighting data, and density plots from the published literature) to the projected exposure level for each individual factor and scored each exposure factor by placing five tallies across the four bins according to the magnitude of exposure projected across the entirety of the stock’s current distribution.”

Line 304: Here it is still confusing how the physical variables were aggregated spatially to be used in comparison with the species range maps.

Please see the response immediately above.

Line 344: I don’t believe “certainty” is an appropriate word here (and in the associated results text). The certainty of your vulnerability scores is intrinsically related to the quality of the data and the expertise and objectivity of the expert participants, neither of which seem to be incorporated into these scores. “Score variance” may be more appropriate here.

The term “certainty” has been used in this capacity in other CVAs that used similar methodology (Hare et al. 2016, Spencer et al. 2019, Crozier et al. 2019, Farr et al. 2021, Giddens et al. 2022, McClure et al. 2023). We have added text (Lines 345–347) explaining “In this context, certainty is a measure of the consistency of the original score with the bootstrap score. The bootstrap score provides an estimate of the vulnerability score sensitivity given the variance in the scoring distribution.” 

Fig. 2 seems redundant with Fig. 1

Fig. 2 has been removed and all other figure numbers updated accordingly

Fig. 3: Please vary the scales along the x axis so that we can see the scores for the non-delphinid taxonomic groups

Scales were updated to allow better resolution and the caption was updated to draw attention to the varying scales. (Lines 441–442)

Lines 426-443: This section literally describes the exact numbers shown in Fig. 3. It’s ok to draw readers attention to the most important findings but, as written, much of this text seems redundant and takes up journal space.

We reduced the text in this section to highlight only specific findings for each taxonomic group. (Lines 430–438)

Line 469: Can you add a sentence here, and perhaps in the figure caption, clarifying that the other attributes did not impact vulnerability scores when they were removed from the analysis? The figure almost looks like there is missing data, but I think these attributes just don’t impact vulnerability.

Added text: “No other factors changed vulnerability scores when omitted during the leave-one-out analysis.” (Lines 466–467)

Added text to figure caption: “Only ocean pH (change in mean), dissolved oxygen (change in mean), air temperature (change in mean), and sea surface temperature (change in mean) changed vulnerability scores when omitted.” (Lines 475–477)

Line 485: Are these unformatted citations?

Yes, thank you for identifying this error. Citations have been updated in the text. (Line 489)

Lines 479-486: This section is biased by the taxonomic skew among the stocks represented in this study. For example, since most of the stocks are dolphins, they dominate the results in this section. I think this analysis would make more sense if it was partitioned by the 5 taxonomic groups

We added taxonomic group-specific plots to a new supplementary material section (S5. Taxonomic group attribute and factor scores).

We added the following text: “Migration was the attribute with the highest median score for 3 of the taxonomic groups (delphinid, ziphiid, and other odontocete) while the prey and diet specificity attribute had the highest median score for the mysticete group and habitat specificity had the highest median score for the pinniped group (S5).” (Lines 481–484)

Additionally, we added parenthetical text to describe results by taxonomic group.

Line 578: “most influential” – is this biased to refer to the delphinid taxa?

Yes. We added text to indicate that the most influential attributes varied by taxonomic group and although stocks within this region were dominated by delphinid stocks, some attributes were important across multiple taxonomic groups (Lines 590–594).

Line 630: I would include “entanglement” explicitly, but you could group this with bycatch by saying “bycatch and entanglement” if you prefer.

Changed text to “bycatch and entanglement” (Line 643)

Line 701-702: Do you mean “data quality scores” here? This is unclear

Added text to indicate we are referring to “lower median data quality scores.” (Line 724)

Reviewer #2:

This manuscript describes a climate-variability assessment of numerous marine mammal stocks in three different regions using expert elicitation. The manuscript was well written and clearly the product of a lot of effort by the authors. As I am not overly familiar with the methods used in the paper, I particularly appreciated the use of examples throughout to help clarify the methods and interpretation of the results. My few comments below are minor and easily addressed by the authors. My 'biggest' criticism is that the Figures 1 -3 convey the main findings of the paper but they are not very visually appealing. I think it does the manuscript a disservice, particularly given that this is a fairly broad paper that could be of considerable interest, and that some relatively changes would make these figures more impactful. Overall though a really interesting and comprehensive manuscript that I look forward to seeing published.

Specific comments:

Table 1: Most of these are the same, just being repeated for different variables. To condense, I would suggest having one row each for the change in mean and change in variability for all the variables except circulation and sea level rise, and then separate rows for circulation and sea level rise since the criteria differed for these two variables

Please see the updated Table 1 with the reviewer’s suggestion incorporated

Figure 3: One of your headings is cut off. I would also suggest using different scales across the column, as bars for non-delphinid groups are difficult to see because of differences in the number of stocks.

The heading has been updated and the scales have been adjusted.

Line 245: I would suggest removing the NARW abbreviation here. I realize it comes up again in the discussion, but since there is such a large gap between the first mention and the second mention, and that the overall number of mentions are relatively small, I don't think it adds more confusion to use an abbreviation.

We changed all occurrences of “NARW” to “North Atlantic right whale.”

Figures: It would be nice of there at least one figure that summarized the results in a very visually appearing nature. Perhaps if possible in some sort of map format, although I recognize this may be too challenging given the range differences in some species

We explored visualizing the results in a map-based format however the number of stocks and the diverse geographic boundaries of the stocks made the map difficult to read.

Editor:

We noticed you have some minor occurrence of overlapping text with the following previous publication(s), which needs to be addressed:

https://spo.nmfs.noaa.gov/sites/default/files/TMSPO196_508.pdf

Overlapping text was identified in the Discussion section and rephrased.

In your revision ensure you cite all your sources (including your own works), and quote or rephrase any duplicated text outside the methods section. Further consideration is dependent on these concerns being addressed

The study can be viewed as a special type of meta-analysis. In a typical meta-analysis, each study is treated as one sample whereas the current study treats one researcher as a sample. Of course, treating one study as one sample has a problem; we all know not all studies are created equal. However, treating one reviewer as one sample also has a problem (they are probably not independent; see comments below). I also think the selection of researchers can lead to bias in the results. In the past, I have experience working with marine mammal scientists. Some of them have strong opinions about what they “like” and what they do “not like” about study results. A large amount of subjectivity is unavoidable in scoring (i.e. there is a filter going from data to scores). It would have been more convincing if the scoring was done by experts in population dynamics without much knowledge of marine mammals. I like the part the study also scores the data quality, but there is still a large uncertainty in how those data are interpreted by the researchers. This needs to be acknowledged. The tone of the language in the abstract and discussion suggesting a high level of confidence is not supported by the study results. In general, a large uncertainty in the results of the study is not quantified sufficiently.

The researchers are treated as independent samples. But I doubt they are. If they work closely with each other (or their institution’s missions are the same), they are dependent. One of the interesting analyses may be to separate NOAA researchers and others. I noticed there are many NOAA researchers involved in research. I am wondering if the results are different between the two groups. If not, perhaps, we do not worry about the institutional dependency. One may be able to group the research based on the taxa of their expertise. I also suggest bootstrapping researchers for all species at the same time to investigate the bias of certain researchers. Some tend to score low (or high) for all species. These and other potential dependencies of researchers will influence the overall results (potentially leading to over-confidence as well as bias in the results). Further investigation of dependency might make the work slightly more convincing.

Applications of expert elicitation can often leave questions about influence and bias.

We used a Delphi approach to minimize biases such as groupthink, deference to authority, and halo effect.

We added the following text to the Discussion (Lines 729-739): Applications of expert elicitation can often leave questions about influence and bias [206, 207]. We used a Delphi approach to minimize biases such as groupthink (a form of consensus seeking due to social pressure), deference to authority (social pressure to agree with more senior or more experienced experts), and halo effect (considering scoring criteria beyond those which were provided) that may be more prevalent with other types of expert elicitation [85, 87, 207-209]. Using an approach in which experts first scored individually before later discussing as a group reduced groupthink and deference to authority by allowing each expert to establish their own score independent of others. Any scoring adjustments occurred independently following the group discussions and we reiterated that consensus was not a goal of the discussions. The group discussions contributed to minimizing halo effect by allowing scorers to identify reasoning that may fall outside the guidelines of the criteria.

Additionally, we used pre-defined criteria for scoring life history traits. Some other CVAs have asked experts to provide their opinion on whether a particular life history trait makes the population more or less vulnerable. To reduce some of the subjectivity associated with those approaches, we had experts score based on how well a population meets the criteria in four separately defined bins for each life history trait. This makes the process more repeatable and provides an opportunity to groundtruth expert scores against published values. The use of these defined bins for each life history trait made the domain-specific knowledge (i.e., marine mammal experts) an important criteria for selecting participants, as they already possessed years of experience with the stocks and an understanding of the literature that would have required extensive training to bring anyone lacking familiarity with marine mammals up to the necessary baseline.

To alleviate concerns about expert effects and institutional influence (i.e. NOAA affiliation), we have included a leave-one-out analysis of the scorers as supplementary material (S6. Expert effect on vulnerability scores) and added text to the Results and Discussion sections.

We added the following text to the Abstract (Lines 93–95): “We quantified sources of uncertainty by bootstrapping vulnerability scores, conducting leave-one-out analyses of individual attributes and individual scorers, and through scoring data quality for each attribute.”

We modified the subheader “Determining attribute importance: Leave-one-out analysis” in the Methods section (Lines 379–384):

“Determining attribute importance and expert effect: Leave-one-out analysis.

We calculated sensitivity and vulnerability scores for each stock by sequentially omitting each sensitivity attribute. We reported the influence of each sensitivity attribute as the change in sensitivity score and vulnerability score by omitting that sensitivity attribute. We conducted a similar analysis for the effect each expert had on vulnerability scores by sequentially omitting the scores of each expert and recalculating each stocks’ vulnerability score.”

We added the following subheader to the Results section (Lines 500–505): “Expert effect

The combination of expert scoring assignments set up 383 scenarios for expert leave-one-out analysis. The effect of removing an individual expert’s scores resulted in no change in vulnerability score in 77% (n=295) of scenarios, a change in vulnerability score of one category (i.e., moving to an adjacent category) in 21% (n=80) of cases, and a change in score of two categories (e.g., moving from low to high) in 2% (n=8) of scenarios (see S6).”

We added the following text to the Discussion section (Lines 700–707): “We conducted a similar leave-one-out analysis by sequentially calculating the vulnerability scores with a single expert’s scores removed to estimate the effect each expert had on scores. We did not encourage experts to work toward consensus during the scoring discussion portion of the process, and variability between experts was expected. One approach to reduce the effect of individual scorers would be to increase the number of scorers per stock. The number of scorers per stock is limited by expert availability, and there are tradeoffs between increasing the number of scorers, increasing individual scorer load, and maintaining a manageable number of scorers for group dynamics and logistical purposes.”

We also highlight the line: “In our study, sources of variability included inter-scorer interpretation of information and variability of underlying experience and knowledge” that appears earlier in the Discussion (Lines 696–697)

---

## [Editor Report · Decision Letter 2]

13 Aug 2023

Vulnerability to climate change of United States marine mammal stocks in the western North Atlantic, Gulf of Mexico, and Caribbean

PONE-D-23-05946R2

Dear Dr. Lettrich,

We’re pleased to inform you that your manuscript has been judged scientifically suitable for publication and will be formally accepted for publication once it meets all outstanding technical requirements.

Kind regards,

Masami Fujiwara, PhD

Academic Editor

PLOS ONE
---

## [Editor Report · Acceptance letter]

17 Aug 2023

PONE-D-23-05946R2 

Vulnerability to climate change of United States marine mammal stocks in the western North Atlantic, Gulf of Mexico, and Caribbean 

Dear Dr. Lettrich:

I'm pleased to inform you that your manuscript has been deemed suitable for publication in PLOS ONE. Congratulations! Your manuscript is now with our production department. 

Kind regards, 

on behalf of

Dr. Masami Fujiwara 

Academic Editor

PLOS ONE